# FeatUp: A Model-Agnostic Framework for Features at Any Resolution

**Stephanie Fu**[*]
MIT

**Mark Hamilton**[*]
MIT, Microsoft

**Laura Brandt**
MIT

**Axel Feldmann**
MIT

**Zhoutong Zhang**
Adobe Research

**William T. Freeman**
MIT, Google

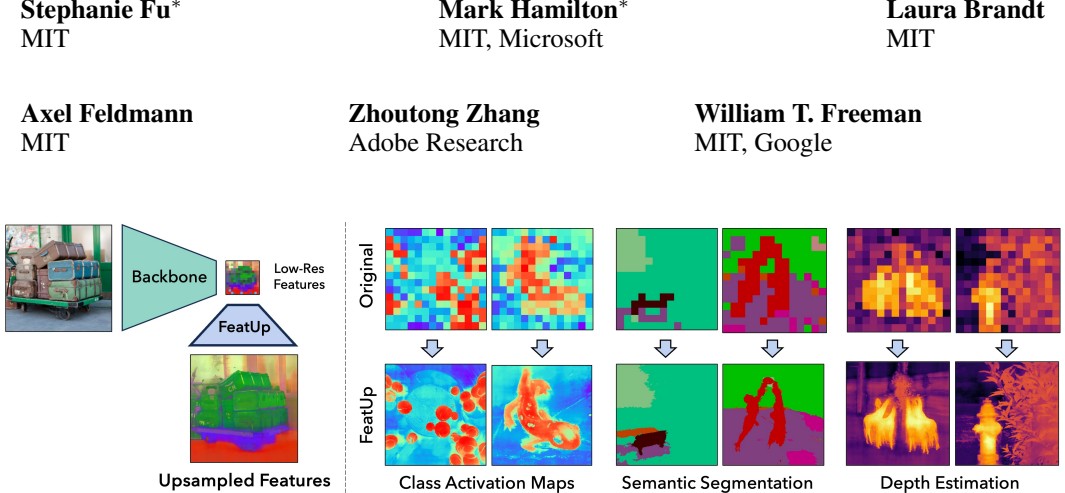

Figure 1: FeatUp upsamples image features from any model backbone, adding spatial resolution to existing semantics. High-res features can be learned either as a per-image implicit network or a general-purpose upsampling operation; the latter is a drop-in module to improve downstream dense prediction tasks.

## Abstract

Deep features are a cornerstone of computer vision research, capturing image semantics and enabling the community to solve downstream tasks even in the zero- or few-shot regime. However, these features often lack the spatial resolution to directly perform dense prediction tasks like segmentation and depth prediction because models aggressively pool information over large areas. In this work, we introduce FeatUp, a task- and model-agnostic framework to restore lost spatial information in deep features. We introduce two variants of FeatUp: one that guides features with high-resolution signal in a single forward pass, and one that fits an implicit model to a single image to reconstruct features at any resolution. Both approaches use a multi-view consistency loss with deep analogies to NeRFs. Our features retain their original semantics and can be swapped into existing applications to yield resolution and performance gains even without re-training. We show that FeatUp significantly outperforms other feature upsampling and image super-resolution approaches in class activation map generation, transfer learning for segmentation and depth prediction, and end-to-end training for semantic segmentation.

## 1 Introduction

Recently, considerable effort has been made to develop methods to extract features from data modalities such as vision (Dalal & Triggs, 2005; LoweDavid, 2004; Weiss et al., 2016; He et al., 2019; Caron et al., 2021), text (Mikolov et al., 2013; Devlin et al., 2018; Radford & Narasimhan, 2018), and audio (Schneider et al., 2019; Hsu et al., 2021). These features often form the backbone of different methods, including classification (Shao et al., 2014), weakly-supervised learning (Ahn et al., 2019; Hamilton et al., 2022), semantic segmentation (Wang et al., 2020), optical flow (Liu et al., 2010; Teed & Deng, 2020), neural rendering (Kobayashi et al., 2022), and more recently, image generation (Rombach et al., 2021). Despite their immense success, deep features often sacrifice spatial resolution for semantic quality. For example, ResNet-50 (He et al., 2015) produces $7 \times 7$ deep features from a $224 \times 224$ pixel input ($32\times$ resolution reduction). Even Vision Transformers (ViTs)

---

[*]Equal contribution. Corresponding authors: markth@mit.edu and fus@csail.mit.edu

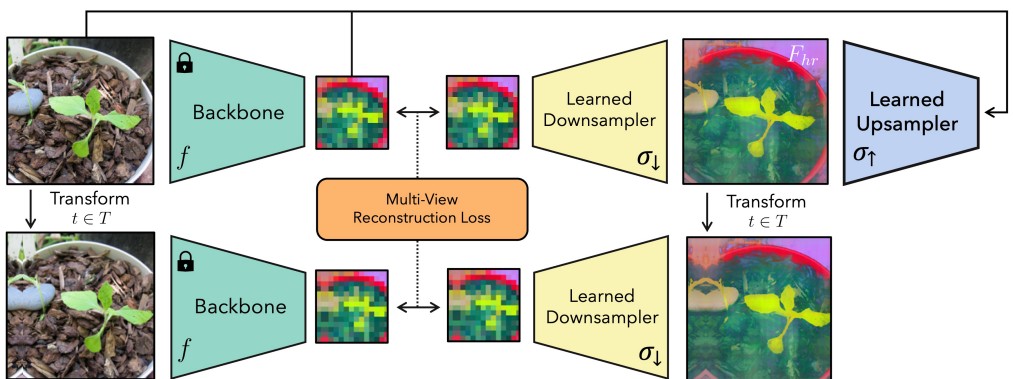

Figure 2: The FeatUp training architecture. FeatUp learns to upsample features through a consistency loss on low resolution "views" of a model's features that arise from slight transformations of the input image.

(Dosovitskiy et al., 2020) incur a significant resolution reduction, making it challenging to perform dense prediction tasks such as segmentation or depth estimation using these features alone.

To mitigate these issues, we propose FeatUp: a novel framework to improve the resolution of any vision model's features without changing their original "meaning" or orientation. Our primary insight, inspired by 3D reconstruction frameworks like NeRF (Mildenhall et al., 2020), is that multiview consistency of low-resolution signals can supervise the construction of high-resolution signals. More specifically, we learn high-resolution information by aggregating low resolution views from a model's outputs across multiple "jittered" (e.g. flipped, padded, cropped) images. We aggregate this information by learning an upsampling network with a multiview consistency loss. Our work explores two architectures for upsampling: a single guided upsampling feedforward network that generalizes across images, and an implicit representation overfit to a single image.

This feedforward upsampler is a parameterized generalization of a Joint Bilateral Upsampling (JBU) filter (Kopf et al., 2007) powered by a CUDA kernel orders of magnitude faster and less memory-intensive than existing implementations. This upsampler can produce high quality features aligned to object edges at a computational cost comparable to a few convolutions. Our implicit upsampler draws a direct parallel to NeRF and overfits a deep implicit network to a signal, allowing for arbitrary resolution features and low storage costs. In both architectures, our upsampled features can be drop-in replacements in downstream applications because our methods do not transform the semantics of the underlying features. We show that these upsampled features can significantly improve a variety of downstream tasks including semantic segmentation and depth prediction. Additionally, we show that model explanation methods such as CAM can be made higher-resolution using upsampled features. In particular, one can study a model's behavior with much greater detail without the need for complex methods based on relevance and information propagation (Lee et al., 2021; Qin et al., 2019). In summary, we include a short video describing FeatUp at aka.ms/featup and make the following contributions:

- FeatUp: a new method to significantly improve the spatial resolution of any model's features, parametrized as either a fast feedforward upsampling network or an implicit network.
- A fast CUDA implementation of Joint Bilateral Upsampling orders of magnitude more efficient than a standard PyTorch implementation and allowing guided upsampling in large-scale models.
- We show that FeatUp features can be used as drop-in replacements for ordinary features to improve performance on dense prediction tasks and model explainability.

## 2  RELATED WORK

**Image-adaptive filtering.**  Adaptive filters are commonly used to enhance images while preserving their underlying structure and content. For example, bilateral filters (Tomasi & Manduchi, 1998; Caraffa et al., 2015; Xiao & Gan, 2012) apply a spatial filter to a low-resolution signal and an intensity filter to a high-resolution guidance to blend information from the two. Joint Bilateral Upsampling (JBU) (Kopf et al., 2007) uses this technique to upsample a low-resolution signal with a high-resolution guidance. JBU has been used successfully for efficient image enhancement and other applications. Recently, some works embed bilateral filtering approaches (Mazzini, 2018) and nonlocal

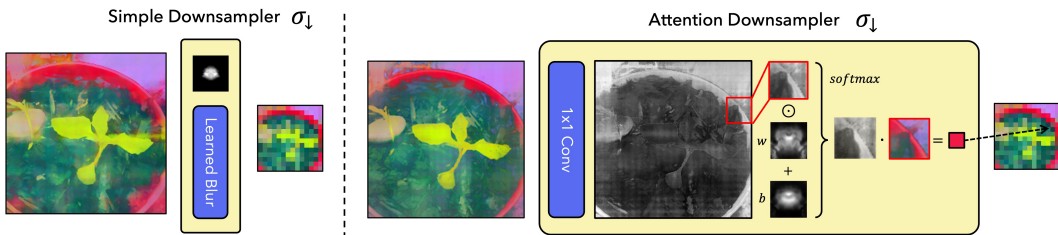

Figure 3: We introduce two learned downsamplers. The simple downsampler (Left) is a fast learned blur kernel. The attention downsampler (right) combines a predicted salience map with spatially invariant kernels. This downsampler can better adapt to networks with nonlinear and dynamic receptive fields.

means (Buades et al., 2005) into convolutional networks (Gadde et al., 2015; Wang et al., 2017) and vision transformers (Qian et al., 2021; Lu et al., 2022d). Shape Recipes (Freeman & Torralba, 2002) learn the local relationship between signals to create up-sample target signals. Pixel-adaptive convolutional (PAC) networks (Su et al., 2019) adapt a convolution operation to input data and has been used to advance performance in segmentation (Araslanov & Roth, 2020; Prangemeier et al., 2020) and monocular depth estimation (Guizilini et al., 2020; Choi et al., 2021; 2020). The Spatially-Adaptive Convolution (SAC) in (Xu et al., 2020) factorizes the adaptive filter into an attention map and convolution kernel. (Gadde et al., 2016) extend bilateral filtering to superpixels and embed this operation inside of a deep network to improve semantic segmentation. This class of methods, effective across a variety of applications, directly incorporates spatial information into the task while still allowing for flexibility in learning a network.

**Image super-resolution.**    One of the earliest deep unsupervised super-resolution methods was Zero-Shot Super-resolution (ZSSR) (Shocher et al., 2018), which learns a single-image network at test time. Local implicit models (Chen et al., 2021) use locally-adaptive models to interpolate information, and have been shown to improve the performance of super-resolution networks. Deep Image Priors (Ulyanov et al., 2020) show that CNNs provide inductive biases for inverse problems such as zero-shot image denoising and super-resolution. While there is extensive literature on image super-resolution, these methods are not well-adapted to handle ultra-low resolution, yet high-dimensional deep features as we show in the Supplement.

**General-purpose feature upsampling.**    A widely-used approach to upsample deep feature maps is bilinear interpolation. Though efficient, this method blurs information and is insensitive to the content or the high-resolution structure in the original image. Nearest neighbor and bicubic interpolation (Keys, 1981) have similar drawbacks. Evaluating a network on larger inputs can achieve higher resolutions but with a steep computational cost. Furthermore, this often degrades model performance and semantics due to the decreased relative receptive field size. For deep convolutional networks, one popular technique is to set final convolution strides to 1 (Long et al., 2015; Qin et al., 2019). However, this approach yields blurry features, as the model's receptive field is still large. Recent works using visual transformers (Amir et al., 2021; Tumanyan et al., 2022) perform a similar modification on input patch strides and interpolate positional encodings. Though simple and reasonably effective, this approach incurs a steep increase in computational footprint for every $2\times$ increase in resolution, making it impossible to use in practice for larger upsampling factors. This approach can also distort features because of the previously mentioned fixed receptive field of the patches.

**Image-adaptive feature upsampling.**    Many different operations exist in the literature to create features at higher resolutions. Deconvolutions (Shi et al., 2016; Dumoulin & Visin, 2016a; Noh et al., 2015; Johnson et al., 2016) and transposed convolutions (Dumoulin & Visin, 2016b) use a learned kernel to transform features into a new space with a larger resolution. The resize-convolution (Odena et al., 2016) appends a learned convolution to deterministic upsampling procedure and reduces checkerboard artifacts that plague deconvolutions (Gauthier, 2015; Odena et al., 2016; Dong et al., 2015). The resize-convolution is now a common component of image decoders such as the U-Net (Ronneberger et al., 2015) and has been applied to semantic segmentation (Li et al., 2018; Huang et al., 2020; Fu et al., 2020) and super-resolution (Lai et al., 2017; Tong et al., 2017; Ledig et al., 2017). Other methods such as IndexNet (Lu et al., 2022a) and Affinity-Aware Upsampling (A2U) (Dai et al., 2020) are effective on image matting but fall short on other dense prediction tasks (Lu et al., 2022b). Methods such as Pixel-Adaptive Convolutions (Su et al., 2019), CARAFE (Wang et al., 2019)SAPA Lu et al. (2022c), and DGF Wu et al. (2019) use learned input-adaptive operators to transform features. Though PAC is flexible, it does not upsample *existing* feature maps faithfully

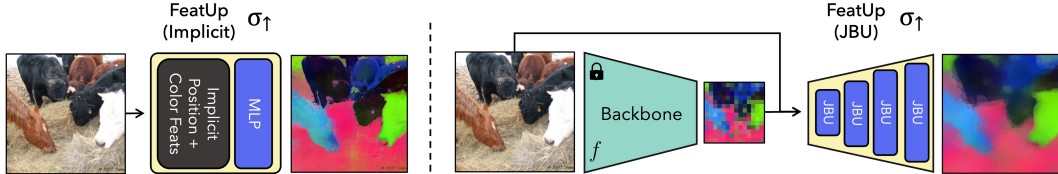

Figure 4: Our Implicit version of FeatUp learns an implicit network to upsample a single image's features. Our JBU FeatUp learns a stack of JBUs that learns to quickly upsample features from a large image corpora.

and instead is used to transform features for downstream tasks. Additionally, DGF approximates the JBU operation with learned pointwise convolutions and linear maps, but does not fully implement JBU because the local query/model is computationally intractable. This is precisely the problem we solve exactly with our new efficient CUDA kernel. Additionally, FADE Lu et al. (2022b) introduces a new semi-shift operator and uses decoder features to produce a joint feature upsampling module. Hu et al. (2022) view feature upsampling in a different light, focusing on a nearest-neighbors approach to align feature maps in encoder-decoder architectures with IFA. While IFA performs well on the specific semantic segmentation benchmarks, it does not take advantage of image guidance and fails to learn high quality representations outside of the encode-decoder framework, as we show in the Supplement.

## 3 METHODS

The core intuition behind FeatUp is that one can compute high-resolution features by observing multiple different "views" of low-resolution features. We draw a comparison with 3D scene reconstruction models such as NeRF (Mildenhall et al., 2020); in the same way that NeRF builds an implicit representation (Sitzmann et al., 2020b; Chen & Zhang, 2019) of a 3D scene by enforcing consistency across many 2D photos of the scene, FeatUp builds an upsampler by enforcing consistency across many low-resolution feature maps. Like in broader NeRF literature, a variety of methods can arise from this basic idea. In this work, we introduce a lightweight, forward-pass upsampler based on Joint Bilateral Upsampling (Kopf et al., 2007) as well as an implicit network based upsampling strategy. The latter is learned per-image and query-able at arbitrary resolution. We provide an overview of the general FeatUp architecture in Figure 2.

The first step in our pipeline is to generate low-resolution feature views to refine into a single high-resolution output. To this end, we perturb the input image with small pads, scales, and horizontal flips and apply the model to each transformed image to extract a collection of low-resolution feature maps. These small image jitters allow us to observe tiny differences in the output features and provide sub-feature information to train the upsampler.

Next, we construct a consistent high-resolution feature map from these views. We postulate that we can learn a latent high-resolution feature map that, when downsampled, reproduces our low-resolution jittered features (see Figure 2). FeatUp's downsampling is a direct analog to ray-marching; just as 3D data is rendered into 2D in this NeRF step, our downsampler transforms high-resolution features into low-resolution features. Unlike NeRF, we do not need to estimate parameters that generate each view. Instead, we track the parameters used to "jitter" each image and apply *the same* transformation to our learned high-resolution features prior to downsampling. We then compare downsampled features to the true model outputs using a gaussian likelihood loss (Hamilton et al., 2020). A good high-resolution feature map should reconstruct the observed features across all the different views.

More formally, let $t \in T$ be from a collection of small transforms such as pads, zooms, crops, horizontal flips, and their compositions. Let $x$ be an input image, $f$ be our model backbone, $\sigma_\downarrow$ be a learned downsampler, and $\sigma_\uparrow$ be a learned upsampler. We can form the predicted high-res features $F_{hr}$ by evaluating $F_{hr} = \sigma_\uparrow(f(x), x)$. We note that this parameterization allows $\sigma_\uparrow$ to be a guided upsampler (which depends on both $x$ and $f(x)$), an unguided upsampler (which depends on only $f(x)$), an implicit network (which depends on only $x$), or a learned buffer of features (which depends on nothing). We can now form our main multi-view reconstruction loss term as follows:

$$\mathcal{L}_{rec} = \frac{1}{|T|} \sum_{t \in T} \frac{1}{2s^2} \| f(t(x)) - \sigma_\downarrow(t(F_{hr})) \|_2^2 + \log(s) \qquad (1)$$

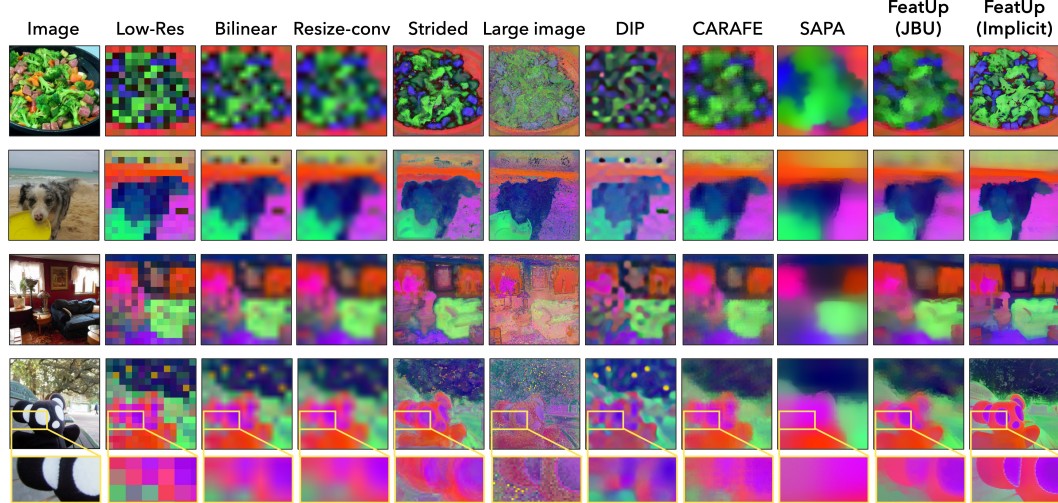

Figure 5: Low-res ViT features ($14 \times 14$) from the COCO-Stuff validation set are upsampled by $16\times$. Bilinear and resize-conv baselines produce blurry outputs. Larger inputs and smaller transformer strides can help, but introduce noise or blur and are bound by time and memory constraints (We can only compute $8\times$ upsamplings for these methods, see Figure 17). Our FeatUp methods preserve semantics of the low-res features and recover lost spatial information from the high-res input image.

Where $\|\cdot\|$ is the standard squared $l_2$ norm and $s = \mathcal{N}(f\,(t\,(x)))$ is a spatially-varying adaptive uncertainty (Hamilton et al., 2020) parameterized by a small linear network $\mathcal{N}$. This turns the MSE loss into a proper likelihood capable of handling uncertainty. This extra flexibility allows the network to learn when certain outlier features fundamentally cannot be upsampled. In the supplement, we show this adaptive uncertainty's effectiveness in an ablation study and visualization.

### 3.1 CHOOSING A DOWNSAMPLER

Our next architectural choice is the learned downsampler $\sigma_\downarrow$. We introduce two options: a fast and simple learned blur kernel, and a more flexible attention-based downsampler. Both proposed modules do not change the "space" or "semantics" of the features with nontrivial transformations, but rather only interpolate features within a small neighborhood. We diagram both choices in Figure 3 and demonstrate the effectiveness of the attention downsampler in Figure 9 of the Supplement.

Our simple downsampler blurs the features with a learned blur kernel and can be implemented as a convolution applied independently to each channel. The learned kernel is normalized to be non-negative and sum to 1 to ensure the features remain in the same space.

Though this blur-based downsampler is efficient, it cannot capture dynamic receptive fields, object salience, or other nonlinear effects. To this end, we also introduce a more flexible attention downsampler that spatially adapts the downsampling kernel. In short, this component uses a 1x1 convolution to predict a saliency map from the high-resolution features. It combines this saliency map with learned spatially-invariant weight and bias kernels and normalizes the result to create a spatially-varying blur kernel that interpolates the features. More formally:

$$\sigma_\downarrow(F_{hr})_{ij} = \text{softmax}(w \odot \text{Conv}(F_{hr}[\Omega_{ij}]) + b) \cdot F_{hr}[\Omega_{ij}] \tag{2}$$

Where $\sigma_\downarrow(F)_{ij}$ is the $i,j$th component of the resulting feature map and $F_{hr}[\Omega_{ij}]$ refers to a patch of high resolution features corresponding to the $i, j$ location in the downsampled features. $\odot$ and $\cdot$ refer to the elementwise and inner products respectively, and $w$ and $b$ are learned weight and bias kernels shared across all patches. Our main hyperparameter for both downsamplers is the kernel size, which should be larger for models with larger receptive fields such as convolutional nets. We defer discussion of model-specific hyperparameters to the Supplement.

### 3.2 CHOOSING AN UPSAMPLER

A central choice in our architecture is the parameterization of $\sigma_\uparrow$. We introduce two variants: "JBU" FeatUp parameterizes $\sigma_\uparrow$ with a guided upsampler based on a stack of Joint Bilateral Upsamplers

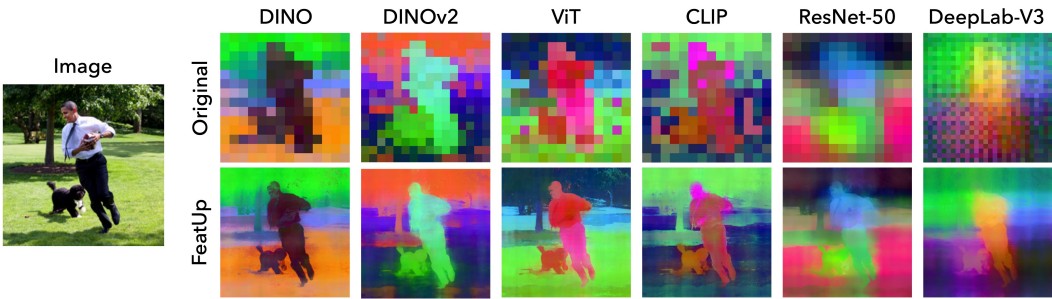

Figure 6: FeatUp can upsample the features of any backbone, even convnets with aggressive nonlinear pooling.

(JBU) (Kopf et al., 2007). This architecture learns an upsampling strategy that generalizes across a corpus of images. The second method, "Implicit" FeatUp, uses an implicit network to parameterize $\sigma_\uparrow$ and can yield remarkably crisp features when overfit to a single image. Both methods are trained using the same broader architecture and loss. We illustrate both strategies in Figure 4.

**Joint Bilateral Upsampler.** Our feedforward upsampler uses a stack of parameterized joint bilateral upsamplers (JBU) (Kopf et al., 2007):

$$F_{hr} = (\text{JBU}(\cdot, x) \circ \text{JBU}(\cdot, x) \circ ...)(f(x)) \tag{3}$$

where $\circ$ is function composition, $f(x)$ is the low-resolution feature map, and $x$ is the original image. This architecture is fast, directly incorporates high-frequency details from the input image into the upsampling process, and is independent of the architecture of $f$. Our formulation generalizes the original JBU (Kopf et al., 2007) implementation to high-dimensional signals and makes this operation learnable. In joint bilateral upsampling we use a high-resolution signal, $G$, as guidance for the low-resolution features $F_{lr}$. We let $\Omega$ be a neighborhood of each pixel in the guidance. In practice, we use a $3 \times 3$ square centered at each pixel. Let $k(\cdot, \cdot)$ be a similarity kernel that measures how "close" two vectors are. We can then form our joint bilateral filter:

$$\hat{F}_{hr}[i,j] = \frac{1}{Z} \sum_{(a,b) \in \Omega} \left( F_{lr}[a,b] \ k_{range}(G[i,j], G[a,b]) \ k_{spatial}([i,j], [a,b]) \right) \tag{4}$$

where $Z$ is a normalization factor to ensure the kernel sums to 1. Here, $k_{spatial}$ is a learnable Gaussian kernel on the Euclidean distance between coordinate vectors of width $\sigma_{spatial}$:

$$k_{spatial}(x,y) = \exp\left( \frac{-\|x-y\|_2^2}{2\sigma_{spatial}^2} \right) \tag{5}$$

Furthermore, $k_{range}$ is a temperature-weighted softmax (Hamilton et al., 2020) applied to the inner products from a multi-layer perceptron (MLP) that operates on the guidance signal $G$:

$$k_{range}(x,y) = \text{softmax}_{(a,b) \in \Omega} \left( \frac{1}{\sigma_{range}^2} MLP(G[i,j]) \cdot MLP(G[a,b]) \right) \tag{6}$$

where $\sigma_{range}^2$ acts as the temperature. We note that the original JBU uses a fixed Gaussian kernel on the guidance signal, $G$. Our generalization performs much better as the MLP can be learned from data to create a better upsampler. In our experiments, we use a two-layer GeLU (Hendrycks & Gimpel, 2016) MLP with 30-dimensional hidden and output vectors. To evaluate $F_{lr}[a,b]$ we follow the original JBU formulation and use bilinear-interpolated features if the guidance pixel does not directly align with a low-resolution feature. For resolution independence, we use coordinate distances normalized to $[-1, 1]$ in the spatial kernel.

One challenge we faced was the poor speed and memory performance of existing JBU implementations. This could explain why this simple approach is not used more widely. To this end, we contribute an efficient CUDA implementation of the spatially adaptive kernel used in the JBU. Compared to a naive PyTorch implementation with the `torch.nn.Unfold` operator, our operation uses up to two orders of magnitude less memory and speeds inference by up to $10\times$. We demonstrate its significant performance improvements in Table 6 of the supplement.

|  | CAM Score | | Semantic Seg. | | Depth Estimation | |
| --- | --- | --- | --- | --- | --- | --- |
|  | A.D. ↓ | A.I. ↑ | Acc. ↑ | mIoU ↑ | RMSE ↓ | $\delta > 1.25$ ↑ |
| Low-res | 10.69 | 4.81 | 65.17 | 40.65 | 1.25 | 0.894 |
| Bilinear | 10.24 | 4.91 | 66.95 | 42.40 | 1.19 | 0.910 |
| Resize-conv | 11.02 | 4.95 | 67.72 | 42.95 | 1.14 | 0.917 |
| DIP | 10.57 | 5.16 | 63.78 | 39.86 | 1.19 | 0.907 |
| Strided | 11.48 | 4.97 | 64.44 | 40.54 | 2.62 | 0.900 |
| Large image | 13.66 | 3.95 | 58.98 | 36.44 | 2.33 | 0.896 |
| CARAFE | 10.24 | 4.96 | 67.1 | 42.39 | 1.09 | 0.920 |
| SAPA | 10.62 | 4.85 | 65.69 | 41.17 | 1.19 | 0.917 |
| FeatUp (JBU) | 9.83 | 5.24 | 68.77 | 43.41 | 1.09 | **0.938** |
| FeatUp (Implicit) | **8.84** | **5.60** | **71.58** | **47.37** | **1.04** | 0.927 |

Table 1: Comparison of feature upsamplers across metrics on CAM faithfulness, linear probe semantic segmentation, and linear probe depth estimation. Both FeatUp variants consistently outperform other approaches, including other forward-pass upsamplers (CARAFE, SAPA) and features optimized at inference-time (DIP).

**Implicit**  Our second upsampler architecture draws a direct analogy with NeRF by parametrizing the high-resolution features of a single image with an implicit function $F_{hr} = \text{MLP}(z)$. Several existing upsampling solutions also take this inference-time training approach, including DIP (Ulyanov et al., 2020) and LIIF (Chen et al., 2021). We use a small MLP to map image coordinates and intensities to a high-dimensional feature for the given location. We follow the guidance of prior works (Mildenhall et al., 2020; Sitzmann et al., 2020a; Tancik et al., 2020) and use Fourier features to improve the spatial resolution of our implicit representations. In addition to standard Fourier positional features, we show that adding Fourier color features allows the network to use high-frequency color information from the original image. This significantly speeds convergence and enables graceful use of high-resolution image information without techniques like Conditional Random Fields (CRFs). We illustrate the profound effect of Fourier color features in Section 6.4 of the Supplement.

More formally, let $h(z, \hat{\omega})$ represent the component-wise discrete Fourier transform of an input signal $z$, with a vector of frequencies $\hat{\omega}$. Let $e_i$ and $e_j$ represent the two-dimensional pixel coordinate fields ranging in the interval $[-1, 1]$. Let : represent concatenation along the channel dimension. We can now express our high-resolution feature map as:

$$F_{hr} = \text{MLP}(h(e_i : e_j : x, \hat{\omega})) \tag{7}$$

Our MLP is a small 3-layer ReLU (Glorot et al., 2011) network with dropout (Srivastava et al., 2014)($p = .1$) and layer normalization (Ba et al., 2016). We note that, at test time, we can query the pixel coordinate field to yield features $F_{hr}$ at **any** resolution. The number of parameters in our implicit representation is over two orders of magnitude smaller than a $(224 \times 224)$ explicit representation while being more expressive, significantly reducing convergence time and storage size.

### 3.3 Additional Method Details

**Accelerated Training with Feature Compression**  To reduce the memory footprint and further speed up the training of FeatUp's implicit network, we first compress the spatially-varying features to their top $k = 128$ principal components. This operation is approximately lossless as the top 128 components explain $\sim 96\%$ of the variance across a single image's features. This improves training time by a factor of $60\times$ for ResNet-50, reduces the memory footprint, enables larger batches, and does not have any observable effect on learned feature quality. When training the JBU upsampler, we sample random projection matrices in each batch to avoid computing PCA in the inner loop. This achieves the same effect thanks to the Johnson–Lindenstrauss lemma (Johnson et al., 1986).

**Total Variation Prior**  To avoid spurious noise in the high resolution features, we add a small ($\lambda_{tv} = 0.05$) total variation smoothness prior (Rudin et al., 1992) on the implicit feature magnitudes:

$$\mathcal{L}_{tv} = \sum_{i,j} \left( (||F_{hr}[i,j]|| - ||F_{hr}[i-1,j]||)^2 + (||F_{hr}[i,j]|| - ||F_{hr}[i,j-1]||)^2 \right) \tag{8}$$

This is faster than regularizing full features and avoids overprescribing how the individual components should organize. We do not use this in the JBU upsampler because it does not suffer from overfitting. We demonstrate the importance of this regularizer in Section 6.4 in the supplement.

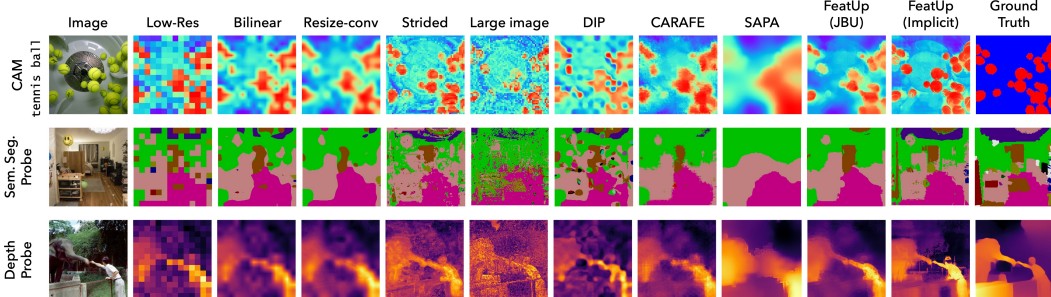

Figure 7: A comparison of different upsampling methods across each of the tasks considered in our analysis. FeatUp achieves significant improvements in resolution across each task.

## 4  EXPERIMENTS

We compare our method against several key upsampling baselines from the literature, in particular: Bilinear upsampling, Resize-conv, Strided (i.e. reducing the stride of the backbone's patch extractor), Large Image (i.e. using a larger input image), CARAFE (Wang et al., 2019), SAPA (Lu et al., 2022c), and FADE (Lu et al., 2022b). We upsample ViT (Dosovitskiy et al., 2020) features by $16\times$ (to the resolution of the input image) with every method except the strided and large-image baselines, which are computationally infeasible above $8\times$ upsampling. For additional details on the strided implementation, please refer to Section 6.2 of the Supplement.

### 4.1  QUALITATIVE COMPARISONS

**Visualizing upsampling methods**   Figure 5 demonstrates the dramatic qualitative improvement FeatUp achieves compared to several baselines. Our visualizations fit a 3-dimensional PCA on each image's low-resolution ViT features and use this PCA to map upsampled features into the same RGB space. We also show that this high-fidelity upsampling extends to higher PCA components in Figure 12, and that FeatUp can improve small object retrieval in Figure 15 in the Supplement.

**Robustness across vision backbones**   Figure 6 demonstrates that FeatUp can upsample a variety of modern vision backbones. In particular, we show the implicit FeatUp features across a variety of backbones spanning transformers, convolutional nets, and both supervised and self-supervised models. Even though backbones like ResNet-50 do not precisely localize objects due to their large receptive fields, FeatUp can reasonably associate features to the correct object.

### 4.2  TRANSFER LEARNING FOR SEMANTIC SEGMENTATION AND DEPTH ESTIMATION

Next, we demonstrate that FeatUp can serve as a drop-in replacement for existing features in downstream applications. To demonstrate this, we adopt the widely used experimental procedure of using linear probe transfer learning to evaluate representation quality. More specifically, we train linear probes on top of low-resolution features for both semantic segmentation and depth estimation. We then freeze and apply these probes to upsampled features to measure performance improvement. If features are valid drop-in improvements, existing probes should work well without adaptation. For all experiments, we use a frozen pre-trained ViT-S/16 as the featurizer, upsample the features (14x14 $\rightarrow$ 224x224), and extract maps by applying a linear layer on the features.

For semantic segmentation, we follow the experimental setting of both (Alain & Bengio, 2016; Hamilton et al., 2022) and train a linear projection to predict the coarse classes of the COCO-Stuff (27 classes) training dataset using a cross-entropy loss. We report mIoU and accuracy on the validation set in Table 1. For depth prediction we train on pseudo-labels from the MiDaS (DPT-Hybrid) (Ranftl et al., 2020) depth estimation network using their scale- and shift-invariant MSE. We report root mean square error (RMSE) and the $\delta > 1.25$ metric which is common in monocular depth estimation literature. More specifically this metric is defined as the percentage of pixels with $\delta = \max(\frac{y}{y^*}, \frac{y^*}{y}) > 1.25$ where $y$ is the depth prediction and $y^*$ is the ground truth.

We stress that these linear probe evaluations show that FeatUp features can improve downstream tasks *without* re-training models. These analyses do not aim to create SOTA segmentation or depth networks. Both FeatUp variants outperform all baselines across all experiments, showing that either variant can be used as a drop-in replacement for existing features. Qualitatively, Figure 7 and Figures 19 - 20 in the supplement show cleaner, more cohesive predictions across both tasks.

| Metric | Bilinear | Resize-conv | IndexNet | A2U | CARAFE | SAPA | FADE | FeatUp (JBU) |
|---|---|---|---|---|---|---|---|---|
| mIoU | 39.7 | 41.1 | 41.5 | 41.5 | 42.4 | 41.6 | 43.6 | **44.2** |
| mAcc | 51.6 | 51.9 | 52.2 | 52.3 | 53.2 | 55.3 | 54.8 | **55.8** |
| aAcc | 78.7 | 79.8 | 80.2 | 79.9 | 80.1 | 79.8 | **80.7** | **80.7** |
| Params (M) | 13.7 | +3.54 | +12.6 | +0.12 | +0.78 | +0.20 | +0.29 | +0.16 |
| GFLOPs | 16.0 | +34.40 | +30.90 | +0.51 | +1.66 | +1.15 | +2.95 | +1.70 |

Table 2: Semantic segmentation results with the Segformer Xie et al. (2021) architecture trained on the ADE20k train set and evaluated on the val set. FeatUp (JBU) outperforms the standard bilinear and resize-conv upsamplers in U-Net architectures, IndexNet Lu et al. (2022a), A2U Dai et al. (2020), and other task-agnostic upsamplers (CARAFE Wang et al. (2019), SAPA Lu et al. (2022c), FADE Lu et al. (2022b)). Additionally, our upsampler is competitive in parameter and floating-point operation count.

### 4.3 CLASS ACTIVATION MAP QUALITY

Attributing a model's predictions to specific pixels is crucial for diagnosing failures and understanding a model's behavior. Unfortunately, common interpretation methods like Class Activation Maps (CAM) are limited by the low res of the deep feature maps and cannot resolve small objects. We show that FeatUp features can be dropped into existing CAM analyses to yield stronger and more precise explanations. More specifically, we use the literature's established metrics, Average Drop (A.D.) and Average Increase (A.I.), that measure CAM quality (refer to Section 6.10 in the Supplement for a detailed description of these metrics). Intuitively, A.D. and A.I. capture how much an image's most salient region changes the classification output. A good CAM should highlight regions with the greatest effect on the classifier's predictions, so censoring these regions will have the largest impact on the model's predictions (lower A.D., higher A.I.). Upsamplers are trained on the ImageNet training set for 2,000 steps, and we compute metrics across 2,000 random images from the validation set. We use a frozen pre-trained ViT-S/16 as the featurizer, and extract CAMs by applying a linear classifier after max-pooling. Upsampling is done (14x14 → 224x224) on the features themselves, and CAMs are obtained from these high-resolution maps. We report results in Table 1, and Figures 7, 18.

### 4.4 END-TO-END SEMANTIC SEGMENTATION

FeatUp not only improves the resolution of pre-trained features but can also improve models learned end-to-end. We adopt the experimental setting of (Lu et al., 2022c;b) to show that our JBU upsampler improves end-to-end performance on ADE20K semantic segmentation using the Segformer (Xie et al., 2021) architecture. Specifically, we train SegFormer on ADE20k Zhou et al. (2019; 2017) (20,210 training and 2,000 val) for 160k steps. To validate that our setup matches that of existing literature despite numerical discrepancies, we also compute FLOPs for SegFormer with various upsamplers in Table 2. These counts are comparable with those in Liu et al. (2023), confirming our architectural setup. We report mean IoU, mean class accuracy (mAcc), and all-pixel accuracy (aAcc) against several recent baselines in Table 2 including IndexNet (Lu et al., 2022a), A2U (Dai et al., 2021), CARAFE (Wang et al., 2019), SAPA (Lu et al., 2022c), and FADE (Lu et al., 2022b) in addition to more standard bilinear and resize-conv operators. Figure 21 in the Supplement shows examples of segmentation predictions across these methods. FeatUp consistently outperforms baselines with fewer added parameters, showing that FeatUp can also improve a broader, jointly trained architecture.

## 5 CONCLUSION

We present FeatUp, a novel approach to upsample deep features using multiview consistency. FeatUp solves a critical problem in computer vision: deep models learn high quality features but at prohibitively low spatial resolutions. Our JBU-based upsampler imposes strong spatial priors to accurately recover lost spatial information with a fast feedforward network based on a novel generalization of Joint Bilateral Upsampling. Our implicit FeatUp can learn high quality features at arbitrary resolutions. Both variants dramatically outperform a wide range of baselines across linear probe transfer learning, model interpretability, and end-to-end semantic segmentation.

## ACKNOWLEDGEMENTS

We would like to thank the Microsoft Research Grand Central Resources team for their gracious help performing the experiments in this work. Special thanks to Oleg Losinets and Lifeng Li for their consistent, gracious, and timely help, debugging, and expertise. Without them, none of the experiments could have been run.

This material is based upon work supported by the National Science Foundation Graduate Research Fellowship under Grant No. 2021323067. Any opinion, findings, and conclusions or recommendations expressed in this material are those of the authors(s) and do not necessarily reflect the views of the National Science Foundation. This research is based upon work supported in part by the Office of the Director of National Intelligence (Intelligence Advanced Research Projects Activity) via 2021-20111000006. The views and conclusions contained herein are those of the authors and should not be interpreted as necessarily representing the official policies, either expressed or implied, of ODNI, IARPA, or the U S Government. The US Government is authorized to reproduce and distribute reprints for governmental purposes notwithstanding any copyright annotation therein. This work is supported by the National Science Foundation under Cooperative Agreement PHY-2019786 (The NSF AI Institute for Artificial Intelligence and Fundamental Interactions, http://iaifi.org/) Research was sponsored by the United States Air Force Research Laboratory and the United States Air Force Artificial Intelligence Accelerator and was accomplished under Cooperative Agreement Number FA8750-19-2- 1000. The views and conclusions contained in this document are those of the authors and should not be interpreted as representing the official policies, either expressed or implied, of the United States Air Force or the U.S. Government. The U.S. Government is authorized to reproduce and distribute reprints for Government purposes notwithstanding any copyright notation herein.

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

## 6 Supplemental Information

### 6.1 Website, Video, and Code

We provide additional details and a short video explaining FeatUp at aka.ms/featup. Additionally, we provide our code at: https://tinyurl.com/28h3yppa

### 6.2 Strided baseline implementation

For the DINO and ViT backbones, we extract patches with a stride of $\frac{16}{\text{upsample factor}}$ to produce a higher density of feature vectors and thus increase feature resolution. We point out that the upsampling factor is limited with this method (as the stride is lower bounded by 1), so this approach can only upsample up to 16x for ViT-S/16. Practically however, these maximum upsampling factors are impractical as they require far more memory than current GPUs provide (see Figure 17).

### 6.3 Comparison to Image-Upsampling Methods

A variety of methods have been proposed for image super-resolution. Among the learning-based approaches, deep image prior (DIP) (Ulyanov et al., 2020) has been used succesfully for enhancing images without additional training data. Figure 8 shows that DIP poorly upsamples features, introducing artifacts and "blob" patterns in the features and downstream outputs. (Shocher et al., 2018) introduced Zero-Shot Super-Resolution, a method that learns an image-specific CNN at test time without additional training data. Additionally, images can be represented as Local Implicit Image Functions (LIIF) (Chen et al., 2021) which can be queried at arbitrary resolution. While similar to FeatUp's implicit network, LIIF trained to continuously represent a feature map does not produce sharp outputs like FeatUp (Figure 8) Despite these methods' successes in the image super-resolution problem space, they are not equipped to upsample high-dimensional features.

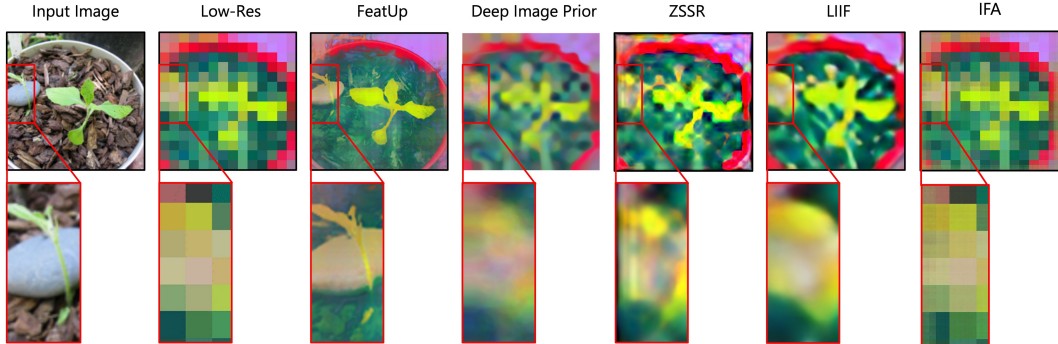

Figure 8: Comparison of image super-resolution methods using Deep Image Prior, Zero-Shot Super-Resolution (ZSSR), and Local Implicit Image Function (LIIF). We also include a visualization on Implicit Feature Alignment (IFA). As shown in the whole feature map and zoomed-in section, thse image upsampling methods do not effectively upsample the low-resolution and high-dimensional feature maps by the large upsampling factors that we are able to handle.

## 6.4 ABLATION STUDIES

We show the effects of each design decision for FeatUp in Figure 9. Our upsampler blurs ResNet features without the uncertainty loss, possibly because it cannot ignore certain nonlinear artifacts or resolve the large pooling window present in ResNet-50. The magnitude regularizer provides smoothing and regularization benefits. Our choice to include Fourier color features dramatically improves resolution and high-frequency details. Finally, the attention downsampler helps the system avoid odd edge and halo effects by learning kernels more focused on salient parts of the signal. Using an explicit buffer of features instead of an implicit network yields significant artifacts, though we note that the artifacts are significantly less dramatic if the simple downsampler is also used.

We also provide an ablation study of the total variation and magnitude regularizers in Figure 11. Our regularizer is fairly robust to different settings as shown by the 2x multiplication for both terms in the 3rd column. However, there still exists an optimal $\lambda$ range that provide important smoothing properties; larger values can interfere with the main reconstruction objective as shown in the final column.

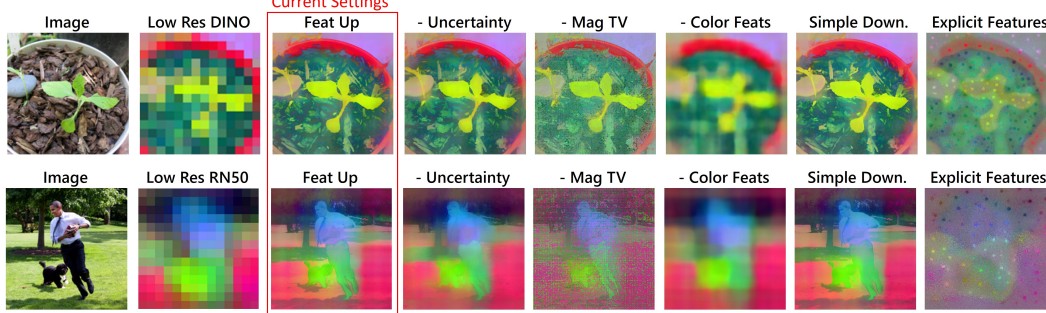

Figure 9: Qualitative ablation study across both DINO and Resnet50 Backbones. The biggest improvements arise from the implicit featurizer, color features, and the magnitude TV regularizer.

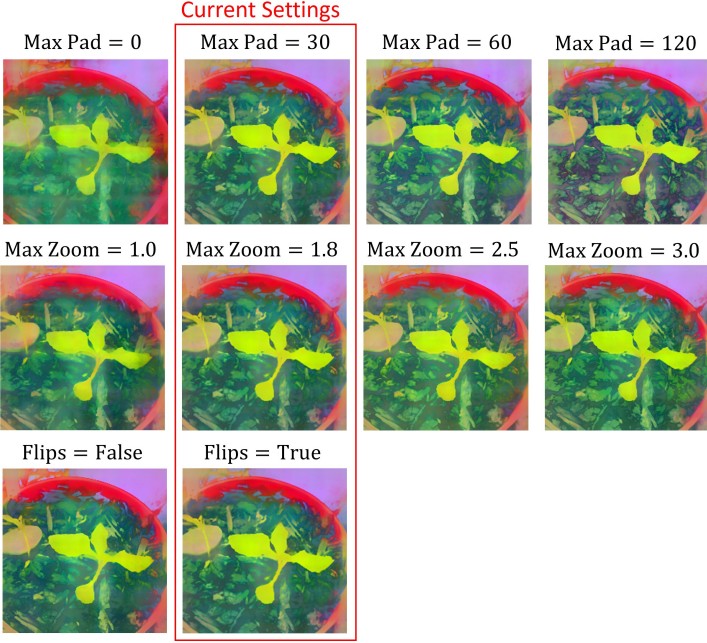

Figure 10: Ablation of FeatUp's training hyper-parameters. We are robust to a range of jitter values, though features degrade with large changes in max pad.

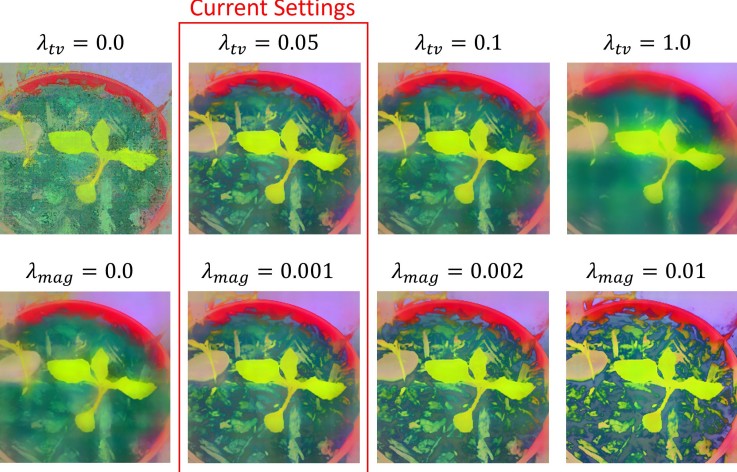

Figure 11: Qualitative ablation study of the TV and magnitude regularizers. FeatUp is fairly robust to the settng of these parameters.

To further justify our design decisions in the context of an end-to-end trained architecture, we evaluate JBU with the Segformer Xie et al. (2021) decoder by 1) removing the MLP (denoted as $MLP$ in Equation 6) on the guidance signal, 2) removing the temperature-weighted softmax and replacing it with Euclidean distance between the central feature and its neighborhood, and 3) removing the softmax and replacing it with cosine distance. Each ablation degrades segmentation performance, with the MLP exclusion being the most detrimental.

| | **FeatUp (JBU)** | | | |
| | Original | - MLP | - Softmax + Euclidean Dist. | - Softmax + Cosine Dist. |
|---|---|---|---|---|
| mIoU | 44.2 | 42.9 | 43.8 | 43.7 |
| mAcc | 55.8 | 54.7 | 54.5 | 55.3 |
| aAcc | 80.7 | 79.4 | 80.0 | 80.4 |

Table 3: Semantic segmentation performance with the Segformer architecture trained on the ADE20k train set and evaluated on the val set. Ablated FeatUp (JBU) replaces the original feature upsampling in the Segformer decoder.

| | CAM Score | | Semantic Seg. | | Depth Estimation | |
| Ablation | A.D. $\downarrow$ | A.I. $\uparrow$ | Acc. $\uparrow$ | mIoU $\uparrow$ | RMSE $\downarrow$ | $\delta > 1.25 \uparrow$ |
|---|---|---|---|---|---|---|
| Original | **9.83** | **5.24** | **68.77** | **43.41** | **1.09** | **0.938** |
| - MLP | 10.04 | 5.10 | 68.12 | 42.99 | 1.14 | 0.917 |
| - Softmax + Euclidean | 9.98 | 5.19 | 68.68 | 43.16 | 1.10 | 0.928 |
| - Softmax + Cosine | 9.97 | 5.21 | 68.49 | 43.15 | 1.12 | 0.924 |

Table 4: FeatUp (JBU) performance with ablated architectural components: removing the MLP, replacing softmax with a gaussian kernel w.r.t. Euclidean or cosine distance. Across all metrics, each ablation degrades performance.

| Attn DS. | O.D. | TV Reg. | CAM Score | | Semantic Seg. | | Depth Estimation | |
|---|---|---|---|---|---|---|---|---|
| | | | A.D. ↓ | A.I. ↑ | Acc. ↑ | mIoU ↑ | RMSE ↓ | $\delta > 1.25$ ↑ |
| ✓ | ✓ | ✓ | **8.84** | **5.60** | **71.58** | **47.37** | **1.04** | **0.927** |
| ✗ | ✓ | ✓ | 9.07 | 5.06 | 70.95 | 46.79 | 1.11 | 0.916 |
| ✓ | ✗ | ✓ | 8.91 | 5.55 | 71.26 | 46.89 | 1.08 | 0.920 |
| ✓ | ✓ | ✗ | 9.10 | 5.00 | 68.06 | 44.36 | 1.11 | 0.913 |

Table 5: Ablation study for implicit FeatUp features with varied downsampler (attention = ✓, simple = ✗), outlier detection, $\lambda_{TV}$ (0.05 = ✓, 0.0 = ✗).

## 6.5 VISUALIZING ADDITIONAL PCA COMPONENTS

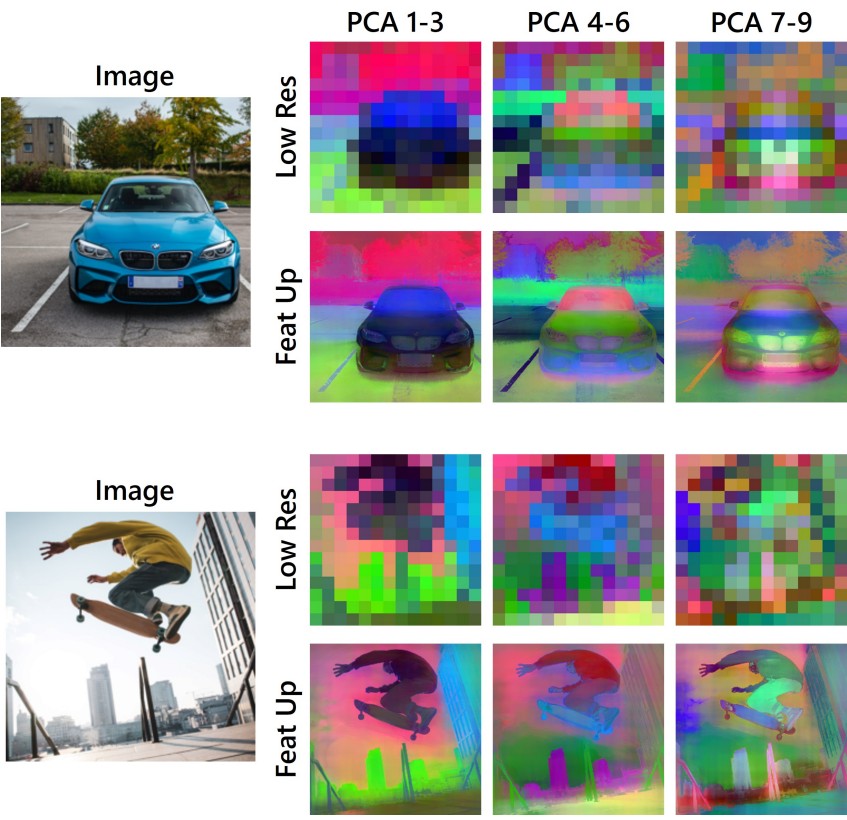

Figure 12: Visualizing higher PCA components with FeatUp. FeatUp upsamples entire feature maps, so their higher-order principal components also remain in the same space as the original features and are upsampled precisely. Higher components tend to separate more fine-grained object categories like the skater from the skateboard, and the trees from the background, and the clouds from the sky. Note that each subobject's features are upsampled precisely to the object it represents.

## 6.6 SALIENCY MAP DETAILS

Downsampling in FeatUp is analogous to ray-marching in NeRF, which approximates the physics of image formation. FeatUp's downsampler approximates a network's process of pooling information into features. As shown in Figure 6, most networks preserve the rough location of objects in their features (the objects just appear downsampled and blurred). This observation leads us to use blur/pooling operators.

The simplest of these is average pooling, but we can do better by generalizing this operation to a learned blur/pooling kernel so the downsampler can better match a network's receptive field size. To map back to NeRF, this is like adding learned camera lens distortion parameters to the ray-marcher so NeRF can better fit the data.

As shown in Figure 6 and described in Section 3.1, even a learned blur/pooling kernel cannot capture dynamic receptive fields or object salience. For example if a small amount of an important object is in a transformer's patch, the whole feature changes. We capture effects like this by making the learned pool/blur kernel dependent on image content using a 1x1 conv (we don't need anything bigger than this layer). This generalizes the previously-described learned blur/pool and allows the downsampler to adaptively pool based on image content. Figure 13 shows that the salience network focuses on certain attributes (e.g. object boundaries, some important small objects). We also note that many common pooling strategies such as average pooling or nearest/bilinear/bicubic resizing are special cases of our learnable attention pooling strategy.

## 6.7 VISUALIZING DOWNSAMPLER SALIENCE AND KERNELS

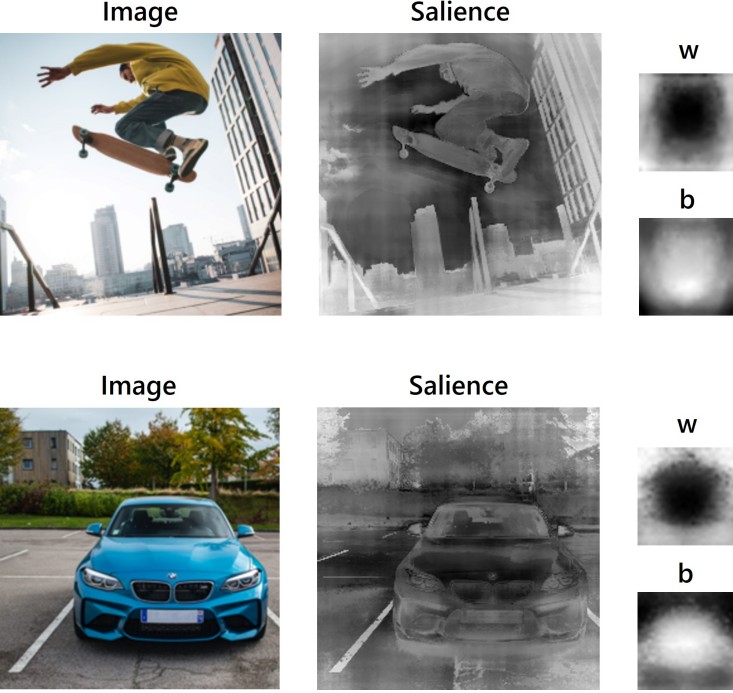

Figure 13: Visualization of downsampler salience and weight and bias kernels for two images. Note how fine-grained objects have higher salience and regions around important objects (like the sky between the hands and the skateboard) have lower salience. This allows the network to capture nonlinear behavior where embeddings from salient regions dominate the embeddings of other regions.

## 6.8 VISUALIZING PREDICTED UNCERTAINTY

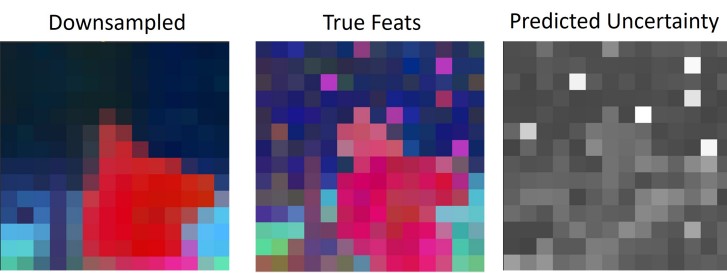

Figure 14: An example predicted uncertainty map for a set of ViT features. White areas have higher uncertainty. In this figure, we can see that nonlinear artifacts like the spurious pink tokens are marked with high uncertainty as they change location depending on the given evaluation. These tokens might serve some other role in the network, such as class-token-like information aggregation. We do not see these types of effects in DINO or convolutional networks.

## 6.9 IMPROVING IMAGE RETRIEVAL FOR SMALL OBJECTS

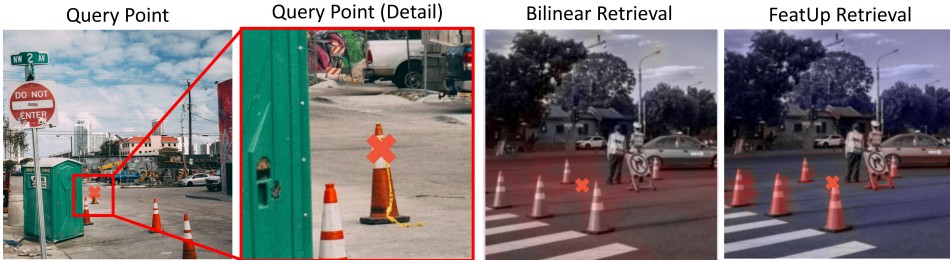

Figure 15: High-resolution FeatUp features can be used to improve the retrieval of small objects and cluttered scenes. A query image (Left) is featurized with DINO and the region marked with a red ×  is used as a query point. We show the detailed placement of this query point in the second image from the left. In the two images on the right, we show the closest matching point in the target image (red ×) and we also visualize the similarity heatmap (red means similarity, blue means dissimilarity). The second image from the right depicts the matching point and heatmap when using bilinear feature interpolation on the image and target. The image on the far right shows the results after upsampling with FeatUp prior to computing the retrieval. Because the scene is cluttered, bilinear interpolation blurs object features together and the resulting query vector attends over both the ground and the traffic cones. FeatUp's features better align with objects allowing only the traffic cones to be retrieved.

### 6.9.1 LINEAR PROBE DETAILS

In both linear probe tasks, one probe was trained on low-resolution (14x14) features from the COCO training set, and frozen for validation across all methods. FeatUp's performance improvements on this repurposed linear probe show that our methods increase resolution without compromising the original feature space. We highlight that these results are *not meant to improve state-of-the-art performance* on segmentation and depth estimation; they are meant to showcase *feature quality* across upsamplers. Because prediction for both tasks is done with a frozen backbone and a single trainable linear probe, the segmentation and depth maps are not meant as a direct application.

### 6.10 AVERAGE DROP AND AVERAGE INCREASE DETAILS

Average Drop is expressed as $\sum_{i=1}^{N} \frac{max(0, Y_i^c - O_i^c)}{Y_i^c} \cdot 100$, where $Y_c^i$ is the classifier's softmax output (i.e. confidence) on sample $i$ for class $c$, and $O_i^c$ is the classifier's softmax output on the CAM-masked sample $i$ for class $c$. We generate $O_i^c$ by keeping the top 50% of CAM values (and Gaussian blurring the remaining 50% of values with less explainability power). Though we generally expect classifiers to drop in confidence because even masking out less-salient pixels can remove important image context, a high-quality CAM will target the explainable regions of an image more precisely and thus maintain a higher confidence. In the reverse direction, we measure the Average Increase to capture the instances where CAM-masked inputs increase model confidence. Specifically, we define Average Increase as $\sum_{i=1}^{N} \frac{\mathbb{1}_{Y_i^c < O_i^c}}{N} \cdot 100$ where $\mathbb{1}_{Y_i^c < O_i^c}$ is an indicator function equal to 1 when $Y_i^c < O_i^c$ - that is, when model confidence increases upon classifying a CAM-masked image.

Similar to the RelevanceCAM evaluation in (Lee et al., 2021), we randomly select 2000 images from the ImageNet validation set (limited to images where the label and model prediction match) to measure A.D. and A.I. on.

### 6.11 PERFORMANCE BENCHMARKING

See Table 6 for performance benchmarking of our adaptive convolution CUDA kernel used in FeatUp (JBU).

| Shape (B, H, W, C, F) | Method | Forward (ms) | Backward (ms) | Peak Mem (Mb) |
|---|---|---|---|---|
| $1 \times 14 \times 14 \times 2048 \times 5$ | Ours | **0.15** | **1.05** | **6.24** |
| | TorchScript | 2455 | 69367 | 12.8 |
| | Unfold | 3.30 | 2.81 | 119. |
| $1 \times 512 \times 512 \times 3 \times 5$ | Ours | **0.55** | **2.10** | **10.2** |
| | TorchScript | 147. | 520. | 24.3 |
| | Unfold | 3.47 | 4.85 | 231. |
| $16 \times 32 \times 32 \times 2048 \times 5$ | Ours | **8.43** | **90.8** | **372.** |
| | Unfold | 118. | 218. | 6628. |
| $32 \times 512 \times 512 \times 3 \times 5$ | Ours | **17.7** | 114. | **326.** |
| | Unfold | 36.0 | **104.** | 4901. |
| $64 \times 14 \times 14 \times 2048 \times 5$ | Ours | **6.12** | **61.1** | **400.** |
| | Unfold | 57.5 | 170. | 5174. |
| $64 \times 224 \times 224 \times 3 \times 5$ | Ours | **6.27** | 36.1 | **128.** |
| | Unfold | 16.7 | **27.4** | 1878. |
| $64 \times 64 \times 64 \times 16 \times 5$ | Ours | **1.06** | **8.99** | **44.5** |
| | Unfold | 7.18 | 14.5 | 822. |
| $64 \times 64 \times 64 \times 16 \times 7$ | Ours | **2.00** | **8.36** | **52.6** |
| | Unfold | 10.8 | 25.6 | 1596. |

Table 6: Comparing the performance of our CUDA JBU kernel with with implementations based on PyTorch's `Unfold` operation and TorchScript. Our implementation dramatically reduces memory overhead and increases inference speed. Code for this operation is available in the provided link.

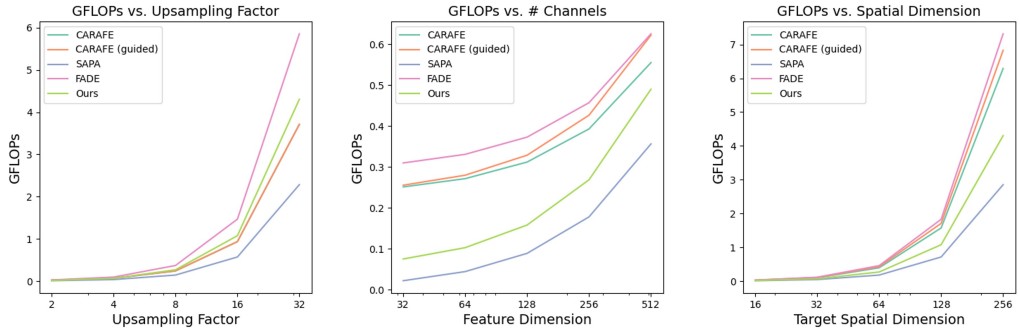

Figure 16: We evaluate how floating point operations scale with various factors. In varying the upsampling factor, feature dimension, and target spatial dimension, FeatUp (JBU) remains competitive in GFLOP usage. For each experiment, the attributes not studied are kept constant (upsampling factor = 2, feature dimension = 256, starting spatial dimension = 8x8).

We analyze peak memory usage and inference time for various upsampling methods. Specifically, we upsample ViT features from a $(1 \times 3 \times 224 \times 224)$ image (i.e. low-resolution feature dimensions of $(1 \times 384 \times 14 \times 14)$) by factors of 2, 4, 8, and 16. Figure 17 shows that FeatUp (JBU)'s peak memory closely follows resize-conv and SAPA baselines and outperforms CARAFE. Additionally, FeatUp is as fast as baselines yet outperforms baselines in all our quantitative evaluations. We note

that strided and large image baselines become computationally infeasible after $8\times$ upsampling, even using a batch size of 1.

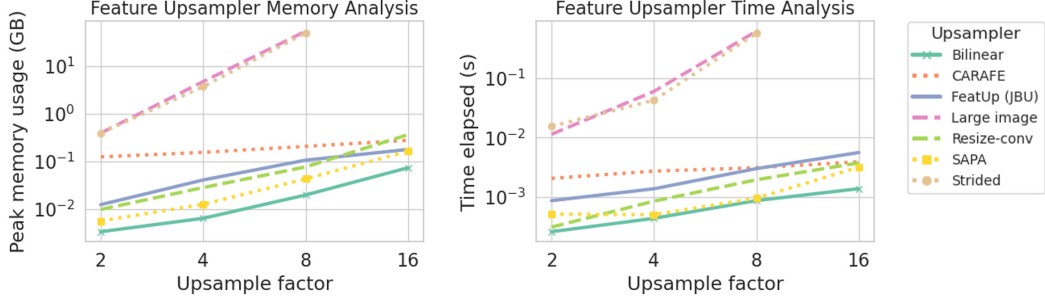

Figure 17: Analysis of peak memory usage (left) and inference time (right) for various forward-pass upsamplers. FeatUp (JBU) is competitive with SAPA and resize-conv across upsampling factors and is more efficient than CARAFE for smaller factors. The large image and strided approaches become infeasible at large upsampling factors we only show metrics for these methods up to $8\times$ upsampling.

## 6.12    ADDITIONAL QUALITATIVE RESULTS

We provide additional CAM visualizations with supervised ViT features on the ImageNet val set in Figure 18. As in the main paper, we upsample features from 14x14 to 224x224 output before extracting CAMs (except for the "Low-Res" column, where the features are kept as-is). Both FeatUp (JBU)'s edge-preserving bilateral filters and the FeatUp (Implicit)'s feature representation allow resulting CAMs to highlight salient regions more accurately. Our CAMs combine the semantic advantages of low-resolution features and the spatial advantages of large images, producing refined versions of the original CAMs without discontinuous patches present in the other upsampling schemes.

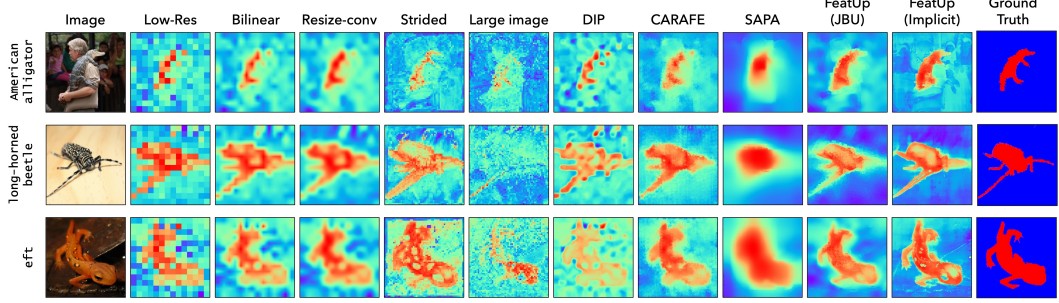

Figure 18: CAMs on the ImageNet validation set from a supervised ViT backbone and linear probe classifier. Both FeatUp variants produce features that are more precise with respect to the input image, allowing downstream CAMs to better align with object boundaries.

See Figure 19 for examples of linear probe transfer learning for semantic segmentation on the COCO-Stuff dataset. The 14x14 features output from a ViT backbone are upsampled with the following methods to achieve 224x224 resolution. Then, a linear probe is trained on the low-resolution features and frozen for evaluation on COCO-Stuff semantic class labels. Our methods recover more cohesive labels of objects and backgrounds.

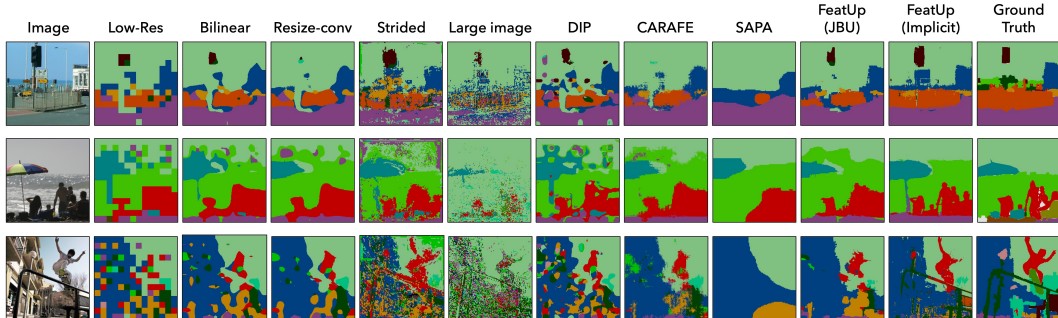

Figure 19: Examples of linear probe transfer learning for semantic segmentation on the COCO-Stuff dataset. Our methods more closely resemble ground-truth segmentation and smooth many of the artifacts present in the low-resolution features. Additionally, FeatUp (Implicit) recovers thin structures like the umbrella pole not even present in the ground truth despite being semantically correct.

Figure 20 provides additional examples of linear probe transfer learning for depth estimation. The 14x14 features output from a ViT backbone are upsampled to achieve 224x224 resolution. Then, a linear probe is trained *directly on the features* to predict depth while supervised with a small MiDaS network. Our results show that both FeatUp variants result in high-quality features capable of transfer learning.

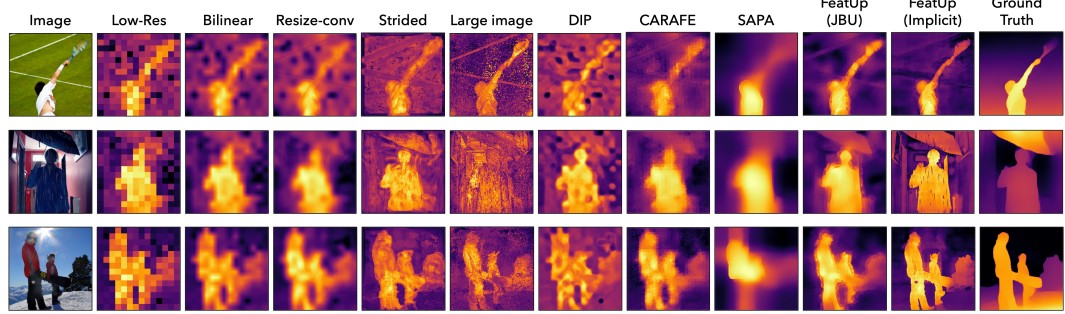

Figure 20: Examples of linear probe transfer learning for depth estimation. Our methods produce sharper object boundaries and smoother interiors that more closely align with true depth than other methods.

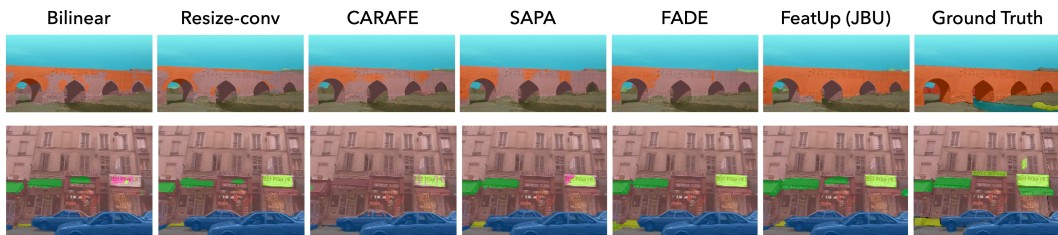

Figure 21: End-to-end training performance of different upsampling methods from our Segformer based semantic segmentation experiments. These results do not use linear probes, but instead train the architecture jointly.

## 6.13 LIMITATIONS

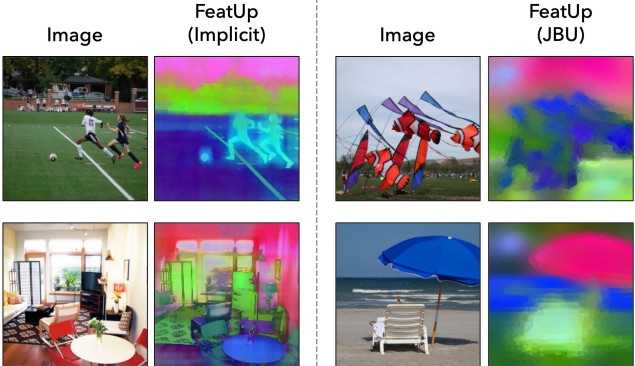

Figure 22: Left: Though FeatUp's implicit network can capture fine detail such as the soccer ball or window frame, it can still produce some halo effects (see soccer player). Additionally, because the method relies on the input image's spatial signal, certain patterns unrelated to object semantics can be transferred to the feature map (see rug pattern), though this is a rare occurrence. Right: FeatUp's JBU network is not as sensitive to fine detail as the implicit network, instead capturing broader contours.

## 6.14 IMPLEMENTATION DETAILS

All backbones (DINO, DINOv2, ViT, ResNet-50, CLIP, and DeepLabV3) used to train FeatUp are frozen, pre-trained models obtained from the community. We outline the hyperparameters used to train FeatUp in table 7.

| Hyperparameter | FeatUp (Implicit) | FeatUp (JBU) |
|---|---|---|
| Num Images | 1 | 4 |
| Num Jitters Per Image | 10 | 2 |
| Downsampler | Attention | Attention |
| Optimizer | NAdam | NAdam |
| Learning Rate | 0.001 | 0.001 |
| Image Load Size | 224 | 224 |
| Projection Dim | 128 | 30 |
| Training Steps | 2000 | 2000 |
| Max Transform Padding | 30px | 30px |
| Max Transform Zoom | $1.8\times$ | $2\times$ |
| Kernel Size | 29 | 16 |
| Total Variation Weight | 0.05 | 0.0 |
| Implicit Net Layers | 3 | n/a |
| Implicit Net Dropout | 0.1 | n/a |
| Implicit Net Activation | ReLU | n/a |

Table 7: Hyperparameters used in training FeatUp.
.

