# OpenReview forum: "FeatUp: A Model-Agnostic Framework for Features at Any Resolution"
_ICLR.cc/2024/Conference — ICLR 2024 poster_

### Official Review · Reviewer_mrZx · 2023-10-29

**Soundness:** 3 good
**Presentation:** 2 fair
**Contribution:** 2 fair
**Rating:** 3
**Confidence:** 5

**Summary:**

This paper presents a model-agnostic framework for feature upsampling. The main contribution of the framework is a multi-view cycle consistency loss where low-res features are matched to its upsampled and downsampled ones. Two instantiations are presented. One is inspired by joint bilateral upsampling, and a fast CUDA implementation is provided. The other is inspired by NeRF where upsampled features are optimized per image. The authors have demonstrated good visualizations of the upsampled feature maps and show that the presented upsampling framework can preserve good semantic smoothness and boundary sharpness. The effectiveness of the framework is validated on a number of tasks.

**Strengths:**

+ Developing general-purpose feature upsampling operators is a recent emerging topic. The goal of designing model-agnostic framework is interesting and shows significantly better visualizations in terms of feature quality over prior art.
+ As far as I can tell, viewing feature upsampling as a rendering problem from multiple low-res views is novel.

**Weaknesses:**

While I generally like the idea of the cycle-consistent training, I have a number of major concerns on the specific operators used/proposed and the results presented.
- **The focus of the paper seems unclear**. Being a feature upsampling framework, the framework should be agnostic to the specific downsampling or upsampling operator chosen. Yet, the current approach leaves me an impression that the framework only works on the proposed learned downsamplers and upsamplers. If the framework matters, some additional existing downsampler-upsampler pairs should be demonstrated to demonstrate its generality; If the proposed operators matter, the contributions and the title may need to be revised.

- Following my previous comment, I think the problem is that the authors attempt to combine many seemingly related techniques together. For instance, the JBU and its fast CUDA implementation look standalone to me. It is unclear to me what the role of the JBU is in the framework, particularly when it is claimed as a contribution. Please explain why JBU is essential to the FeatUp framework. In addition, applying JBU to feature upsampling is not new. "Fast End to End Trainable Guided Filter" CVPR18 and "Superpixel Convolutional Networks using Bilateral Inceptions" ECCV16 are two example works, which is also missing in the related work.

- **The ablation study of the paper is poor**. It is unclear to me what exactly leads to the good feature quality. Aside from the loss, there are a number of tricky designs in the downsampler and upsampler used. For example, why a blur kernel must be used in downsampling? And why introducing the saliency map and how it works? Why a simple 1x1 convolution is sufficient to present the saliency map? Would it be precise enough? Also, the implicit FeatUp produces significantly better visualizations according to Fig. 5. Is it because of the per-image optimization. A thorough ablation study is required.

- **The method may be justified in an inappropriate setting**. According to Table 1, FeatUp (Implicit) has reported significant performance improvements on the transfer learning setting. Yet, this setting is rarely used in feature upsampling literature. Why choosing this setting? I wonder how the parameters of comparing upsamplers are updated in this setting. In addition, the comparison between the implicit variant and other upsamplers seems unfair, because other upsamplers do not optimize the per-image upsampling quality. This also leads to the next issue.

- **The practicality of the implicit FeatUp seems poor**. In reality, the per-image optimization of upsampling is unrealistic, it is not likely that the implicit variant is used as a plug-in in existing networks (the experimental setting in Table 2 also confirms my opinion). But I think this approach can somehow indicate an upper bound of upsampling quality.

- The performance of baseline approaches does not align with the number reported in the corresponding papers and is significantly lower. For example, SAPA reports 44.4 mIoU but the table in Table 2 only reports 41.6. Please check.

**Questions:**

Please see the weaknesses for my major concerns. I have also some other suggestions:

Is it appropriate to indicate augmented low-res features as different views? In 3D, different views often imply different view points.

In addition, the organization of related work does not make sense to me.
- First, the paper does not conduct experiments on image super-resolution. This sub-section seems redundant.
- Second, the references in general-purpose feature upsampling are weird. Both Amir et al. (2021) and Tumanyan et al. (2022) do not study feature upsampling at all. Why are the two papers cited? In my opinion, most references discussed in the section of image-adaptive feature upsampling should be discussed in this section. As far as I know, CARAFE is one of the first paper studying general-purpose feature upsampling.
- In addition, some closely related approaches compared in experiments are not discussed/compared in related work such as FADE and SAPA.

Please address my major concerns in the rebuttal. I would re-evaluaute the paper conditioned on the response.

**Details Of Ethics Concerns:**

The paper studies feature upsampling. In this context, the ethics concerns may not exist.

---

> ### Author Response · Authors · 2023-11-18
>
> We thank the reviewer for their helpful comments.
>
> **Quantitative Ablations**
>
> We added ablations of every major architectural decision of FeatUp in the global response. Each ablation degrades performance across the metrics considered.
>
> **Justifying the focus of the paper**
>
> We present FeatUp as a framework that generalizes across backbones and upsamplers. The JBU and Implicit variants are our novel upsampler contributions that outperform existing upsamplers, but we do evaluate others. In particular:
> - An important aspect of generality is the fact that FeatUp can work with any backbone network (see Fig. 6), making it applicable to many models in the community.
> - Though we present our novel JBU upsampler that improves performance, Table 1 shows 4 other upsamplers can be trained with FeatUp including implicit networks, resize-convs, CARAFE [1], and SAPA [2]. These upsamplers are all trained with our proposed scheme (besides bilinear, strided, and large-image methods, which aren't trained). We note that our JBU variant, however, provides improvements over all other upsamplers.
> - Though we provide 2 example downsamplers, we stress that this component is primarily dictated by “nature”. To draw a direct analogy with NeRF [3], the downsampler is analogous to ray-marching, which approximates the physics of light. In NeRF, this component can be modified slightly, (lens distortion, surface properties, light fields, etc) but needs to approximate physics. For FeatUp, Figure 6 shows that modern networks pool/blur information which leads us towards pooling/blurring operators for the downsampler. We also elaborate on this in the global response section “Clarifying the role of the saliency maps”. We have justified these downsamplers with extensive experiments, visualizations, and ablations.
>
> **Justifying the experimental settings**
> - We use existing experimental settings from the literature (FADE [4] + SAPA) for Table 2
> - We compare against other per-image optimization methods including Deep Image Priors (DIP) [5] in Table 2 and Figure 8 and ZSSR [6] in Figure 8.
> - Transfer learning is an established setting to measure feature quality. It is used throughout the unsupervised representation learning literature including in methods like STEGO [7], PiCIE [8], MOCO [9], SimCLR [10], DINO [11] and others. This analysis also shows we improve performance on downstream tasks *without retraining* downstream components, which is important for practitioners who want to upgrade or better understand existing methods.
> - The parameters in the upsampler comparison of Table 1 are updated using the FeatUp training losses. We use these upsampling operators instead of JBU demonstrating that our framework works with a variety of upsamplers.
>
> **Role of JBU**
>
> Our paper focuses on making a system that can efficiently upsample features and retain their original semantics. We find that existing upsampling layers do not properly utilize guidance information as well as the JBU does. Our analysis shows our JBU operator beats all existing upsamplers for creating upsampled features, which is why we investigated it. We hope you can see this as an additional novelty of our work.
>
> **Explaining our relationship to “Fast End to End Trainable Guided Filter”**
>
> We thank the reviewer for bringing this work to our attention and have added it to our related work section. We note that “Fast End to End Trainable Guided Filter” [12] does not perform the full JBU computation but instead approximates it with learned pointwise convolutions and linear maps because the local query/model was computationally intractable. This is precisely the problem we solve exactly with our new efficient CUDA kernel.
>
> **Justification of FeatUp (Implicit) method**
>
> Please refer “Practicality of FeatUp (Implicit)” in the global response for advantages of any-resolution features. FeatUp (Implicit) is not the only per-image-optimized method as we compare to the widely-used approaches DIP and ZSSR that optimize results per image.
>
> **Performance discrepancy of SAPA**
>
> We appreciate you pointing this out. We have used SAPA’s official code, and use the encoder-decoder architecture and training objective detailed in SegFormer [13] and followed by FADE evaluations. All experiments are performed using the same controlled settings and environment to ensure a fair comparison between methods. We note this discrepancy the paper for full transparency.
>
> **Organization of Related Work**
>
> We have incorporated all suggested works into the related work. We evaluate some image super-resolution methods (DIP, ZSSR, and LIIF) in Section 6.3 which is why we included them in our related work. We shortened this section and directed readers to the supplement to provide more context. Regarding Amir et al. (2021) and Tumanyan et al. (2022), both works describe a modified patch embedding stride which is our “strided” baseline.
>
> **All details and results in this response have been added to our paper.**

---

> > ### Comment · Reviewer_mrZx · 2023-11-22
> > **Thank you for the responses.**
> >
> > I have read the rebuttal. I am sorry for the late reply, mostly because I have re-read the paper and the revision to double check some details. The rebuttal addresses part of my concerns, while some remain (probably cannot be addressed in a rebuttal). Let my detail my further comments.
> >
> > - On the focus of the paper. The authors provide an justification and acknowledge the JBU and Implicit variants are also their upsampler contributions. This confirms my point. As the boundary between a generic upsampling framework and two upsampling operators are blurred. Especially when the framework is mainly used to justify the two upsampling operators. The authors also comment that "Table 1 shows 4 other upsamplers can be trained with FeatUp including implicit networks, resize-convs, CARAFE [1], and SAPA [2]". This confuses me. I have re-read the paper and the revision and find nowhere any text or training details stating how the comparing operators are trained with the proposed framework. In addition, I am also interested to know how a single upsampling operator is trained within the framework where a coupled downsampling-upsampling operator pairs are required. As far as I know, resize-convs, CARAFE, and SAPA are just upsampling operators. Anyway, this is a place, which, I think, needs further clarification.
> >
> > - For the missing ablation study, the authors have supplemented many additional results. They have mostly addressed my points.
> > One further suggestion. To justify the effectiveness and generality of the framework, in addition to the proposed two operators, a better choice may be to validate how existing coupled downsampling and upsampling operators work with the framework such as max pooling-max unpooling and space2depth-depth2space.
> >
> > - The linear-probe experimental setting. As the authors state, the linear-probe setting is mainly used for unsupervised representation learning. If the proposed framework is training-free when used with downsampling tasks, this setting seems fine. But unfortunately, the framework still needs training with existing backbones. In this case, training the upsampling operators with a task-orientated loss and update the network parameters can mostly reveal the upper bound capability of the operators. I think this is the reason why existing work evaluates upsampling under an end-to-end setting as Table 2. Also, I am curious to know how the framework works under the end-to-end setting in Table 2. Does it use the cylic loss and the task loss together? If this is true, the additional loss may be used as well on other comparing operators.
> >
> > - Practicality of FeatUp (Implicit). The rebuttal has solved my concern. The reviewer agrees in some applications the quality is better than the speed.
> >
> > - The performance discrepancy of SAPA. If the authors find a diffculty in reproducing the results, a better way is to cite the results reported by the paper or to raise a issue at the online repository to ask why. Simply reporting an unmatched number seems inappropriate without any grounded explanation.
> >
> > Overall, while I still have some further concerns, I would like to upgrade my rating to Borderline considering the comprehensive rebuttal by the authors (so far I do not have a preference for neither acceptance nor rejection).

---

> ### Author Response · Authors · 2023-11-18
>
> **References**
>
> [1] Wang, Jiaqi et al. “CARAFE: Content-Aware ReAssembly of FEatures.” 2019 IEEE/CVF International Conference on Computer Vision (ICCV) (2019): 3007-3016.
>
> [2] Lu, Hao et al. “SAPA: Similarity-Aware Point Affiliation for Feature Upsampling.” ArXiv abs/2209.12866 (2022): n. pag.
>
> [3] Mildenhall, Ben et al. “NeRF: Representing Scenes as Neural Radiance Fields for View Synthesis.” Commun. ACM 65 (2020): 99-106.
>
> [4] Lu, Hao, et al. "FADE: Fusing the assets of decoder and encoder for task-agnostic upsampling." ECCV 2022.
>
> [5] Ulyanov, Dmitry et al. “Deep Image Prior.” International Journal of Computer Vision 128 (2017): 1867 - 1888.
>
> [6] Shocher, Assaf et al. “Zero-Shot Super-Resolution Using Deep Internal Learning.” 2018 IEEE/CVF Conference on Computer Vision and Pattern Recognition (2017): 3118-3126.
>
> [7] Hamilton, Mark et al. “Unsupervised Semantic Segmentation by Distilling Feature Correspondences.” ArXiv abs/2203.08414 (2022): n. pag.
>
> [8] Cho, Jang Hyun et al. “PiCIE: Unsupervised Semantic Segmentation using Invariance and Equivariance in Clustering.” 2021 IEEE/CVF Conference on Computer Vision and Pattern Recognition (CVPR) (2021): 16789-16799.
>
> [9] He, Kaiming et al. “Momentum Contrast for Unsupervised Visual Representation Learning.” 2020 IEEE/CVF Conference on Computer Vision and Pattern Recognition (CVPR) (2019): 9726-9735.
>
> [10] Chen, Ting et al. “A Simple Framework for Contrastive Learning of Visual Representations.” ArXiv abs/2002.05709 (2020): n. pag.
>
> [11] Caron, Mathilde et al. “Emerging Properties in Self-Supervised Vision Transformers.” 2021 IEEE/CVF International Conference on Computer Vision (ICCV) (2021): 9630-9640.
>
> [12] Wu, Huikai et al. “Fast End-to-End Trainable Guided Filter.” 2018 IEEE/CVF Conference on Computer Vision and Pattern Recognition (2018): 1838-1847.
>
> [13] Xie, Enze, et al. "SegFormer: Simple and efficient design for semantic segmentation with transformers." NeurIPS 2021

---

> ### Author Response · Authors · 2023-11-22
> **Thank you for your response.**
>
> We thank the reviewer for their considerate response and for raising their rating to a 5. Could you please edit your original response to mark the improvement in score so that the system registers the change? We will upload another copy of the full text today that takes this additional round of feedback into account. Regarding your follow up points:
>
> >  Especially when the framework is mainly used to justify the two upsampling operators.
>
> The implicit version of the upsampler is justified moreso by analogy to NeRF and broad use in the literature. The JBU upsampler is also justified through an end to end semantic segmentation experiment, completely separate from the FeatUp losses and broader framework.
>
> >  The authors also comment that "Table 1 shows 4 other upsamplers can be trained with FeatUp including implicit networks, resize-convs, CARAFE [1], and SAPA [2]". This confuses me. I have re-read the paper and the revision and find nowhere any text or training details stating how the comparing operators are trained with the proposed framework. .... As far as I know, resize-convs, CARAFE, and SAPA are just upsampling operators. Anyway, this is a place, which, I think, needs further clarification.
>
> We adjusted the main text to make this training clearer. In the same way that multi-view consistency can be used to learn JBU parameters, we can use this same loss to learn parameters of SAPA, CARAFFE, or Resize Convs which all have trainable parameters that must be learned. Functionally, these upsamplers are identical to our JBU with just differences in their parameterizations and internal details.
>
> > In addition, I am also interested to know how a single upsampling operator is trained within the framework where a coupled downsampling-upsampling operator pairs are required.
>
> Coupled upsamplers and downsamplers are not required. The downsampler acts as the ray-marcher does in NeRF, you can fit any kind of 3d representation (implicit, feedforward, etc) and just using ray-marching to get from 3d to 2d. In FeatUp any downsampler can be trained using either downsampler. Again, the downsampler is just an operation to learn the way deep networks pool semantics and is unrelated to the details of the upsampler. This is why we think of it as an ``independent'' component.
>
> >  To justify the effectiveness and generality of the framework, in addition to the proposed two operators, a better choice may be to validate how existing coupled downsampling and upsampling operators work with the framework such as max pooling-max unpooling and space2depth-depth2space.
>
> It is important to note that the upsampler and downsampler don’t need to be paired in FeatUp. Its more important to have a well parameterized downsampler so that it can capture the nonlinear and dynamic pooling behavior of a variety of modern deep networks. If after reading this, you still feel that the work needs these experiments we can start working on them.
>
>
> > If the proposed framework is training-free when used with downsampling tasks, this setting seems fine. But unfortunately, the framework still needs training with existing backbones. In this case, training the upsampling operators with a task-orientated loss and update the network parameters can mostly reveal the upper bound capability of the operators. I think this is the reason why existing work evaluates upsampling under an end-to-end setting as Table 2.
>
> The important take away from table 1 is that you can directly improve the resolution of existing deep models without changing anything important about their deep features other than the resolution. This is critical if you want to use the high-res features of feat up without retraining other components of an architecture. It is also important if you want to study the behavior of existing models through CAM. The linear probe results also show that this resolution improvement fundamentally improves the quality of existing features as evaluated by the standard evaluation method used in hundreds of papers in the literature.
>
> >Also, I am curious to know how the framework works under the end-to-end setting in Table 2. Does it use the cylic loss and the task loss together? If this is true, the additional loss may be used as well on other comparing operators.
>
> For the sake of making apples-to-apples comparisons we do not use our FeatUp losses to train the models in table 2, we only use the task loss.
>
> >The performance discrepancy of SAPA. If the authors find a diffculty in reproducing the results, a better way is to cite the results reported by the paper or to raise a issue at the online repository to ask why. Simply reporting an unmatched number seems inappropriate without any grounded explanation.
>
> We will include both numbers and put an asterisk next to the literature reported one with an expanded description. We will also try to get in touch with the SAPA authors.

---

### Official Review · Reviewer_2f29 · 2023-11-01

**Soundness:** 4 excellent
**Presentation:** 4 excellent
**Contribution:** 3 good
**Rating:** 6
**Confidence:** 4

**Summary:**

This paper introduces a method FeatUp that learns to upsample a low-resolution feature map at any-resolution. The proposed method is supervised by multiview consistency, and has two variants, one with a single forward and one fits an implicit model per image. The authors compare the method to several prior baselines for feature upsampling.

**Strengths:**

1. Upsampling deep feature maps of neural networks is an important research topic and has wide applications.
2. The idea of connecting feature upsampling to implicit neural representations and using multiview consistency to supervise the upsampled results is interesting and intuitive.
3. The effectiveness of the proposed method is well visualized in several figures.

**Weaknesses:**

1. A main goal of upsampling the feature map is to use it for tasks that requires semantic understanding and dense prediction, e.g. semantic segmentation. How does the proposed method work for more standard semantic segmentation benchmarks (for both mIoU and computation cost) and what are the main advantages of the proposed method? For example, in the context of any-resolution upsampling of feature map, Learning implicit feature alignment function for semantic segmentation (ECCV 2022) seems to be related, does the proposed method show advantages over the prior work?
2. How does the computation cost (in FLOPS) compare to prior works in main tables? Since the proposed method is a new upsampler, it would better demonstrate the advantages of the proposed method by comparing the computation cost.
3. In practice, why the reconstruction of features at any-resolution is important? Typically, the encoder takes a fixed-resolution input, the task is more likely to be fixed-resolution rather than any-resolution. If any-resolution is not the key point, can the proposed method improve over state-of-the-art semantic segmentation methods on standard benchmarks?

**Questions:**

Why a predicted salience map is needed in attention downsampler?

---

> ### Author Response · Authors · 2023-11-18
>
> We thank the reviewer for their helpful comments. We address questions and concerns below.
>
> **Segmentation Benchmarks**
>
> We evaluate commonly-used upsamplers on the SegFormer [1] architecture and the ADE20k [2] dataset, which is a standard semantic segmentation benchmark. As seen in Table 2, FeatUp outperforms the other upsamplers in mIoU, mAcc, and aAcc while remaining competitive in parameter count (it uses fewer parameters than every upsampler meant for features, with only A2U being lighter - however, A2U was designed for image matting and does not perform well on this task).
>
> **Comparison to additional references**
>
> We thank the reviewer for pointing out related prior work on feature upsampling, and have added them in our related work section.
>
> We have also implemented “Learning Implicit Feature Alignment Function for Semantic Segmentation” [3], and added a visualization to Figure 8. While IFA performs well on specific semantic segmentation benchmarks, it does not take advantage of image guidance and fails to learn high quality representations outside of the encode-decoder framework.
>
> **Computation Cost Analysis**
>
> We analyze how GFLOPs (= FLOPs * 1e-9) scale for each upsampling operation in a new figure (see Figure 16 in the paper) and we add an analysis of FLOPS to Table 2. In general we find that FeatUp’s JBU operator is lightweight compared to other upsamplers and is small compared to the cost of evaluating the backbone network.
>
> In particular, we study the original CARAFE [4], a “guided” CARAFE (the original operation with the guidance concatenated with the source to mirror the other guided operations in this study), SAPA [5], FADE [6], and FeatUp (JBU). Besides the attribute being varied (upsampling factor, feature dimension, or target spatial dimension), all other experimental settings are kept constant and exemplify a realistic feature-upsampling scenario (detailed below).
>
> - First, we vary the upsampling factor. In each experiment, we begin with a $(1, 256, 8, 8)$ tensor, guide it with a $(1, 256, 8n, 8n)$ tensor when guidance is appropriate, and produce a $(1, 256, 8n, 8n)$ tensor with $n \in \{ 2, 4, 8, 16, 32 \}$.
> - In the second plot, we vary the channel dimension. In each setup, we begin with a $(1, c, 8, 8)$ tensor, guide it with a $(1, c, 16, 16)$ tensor when guidance is appropriate, and produce a $(1, c, 16, 16)$ tensor with $c \in \{ 32, 64, 128, 256, 512 \}$.
> - Finally, we vary the spatial dimension of the output. In each setup, we begin with a $(1, 256, s//2, s//2)$ tensor, guide it with a $(1, 256, s, s)$ tensor when guidance is appropriate, and produce a $(1, 256, s, s)$ tensor with $c \in \{ 16, 32, 64, 128, 256 \}$.
>
> Note that the resize-conv operation’s GFLOPs can also be calculated in the same way, but when visualized on these plots, its computational load grows so quickly that the upsamplers depicted appear to have constant performance. Thus, we show the upsamplers that are most competitive in performance.
>
> We find that FeatUp is competitive in GFLOP count to the current leading upsamplers and is lighter weight than FADE. It is comparable to CARAFE’s computational cost. These GFLOP analyses show that FeatUp (JBU) is not only architecturally and qualitatively a reasonable substitute for existing upsamplers, it is competitive in # FLOPs across a wide array of scenarios.
>
> **Importance of Any-Resolution Features**
>
> Please refer to the global response (“Practicality of FeatUp (Implicit)”) for descriptions of use cases for any-resolution features.
>
> **Importance of the Predicted Salience Map**
>
> Please refer to the global response (“Clarifying the role of the saliency map”) for an explanation of this design decision.
>
> **All details and new results in this response have been added to our paper revision (in yellow highlighted text).**
>
> **References**
>
> [1] Xie, Enze, et al. "SegFormer: Simple and efficient design for semantic segmentation with transformers." Advances in Neural Information Processing Systems 34 (2021): 12077-12090.
>
> [2] Zhou, B., Zhao, H., Puig, X., Xiao, T., Fidler, S., Barriuso, A., & Torralba, A. (2019). Semantic understanding of scenes through the ade20k dataset. International Journal of Computer Vision, 127(3), 302-321.
>
> [3] Hu, Hanzhe et al. “Learning Implicit Feature Alignment Function for Semantic Segmentation.” European Conference on Computer Vision (2022).
>
> [4] Wang, Jiaqi et al. “CARAFE: Content-Aware ReAssembly of FEatures.” 2019 IEEE/CVF International Conference on Computer Vision (ICCV) (2019): 3007-3016.
>
> [5] Lu, Hao et al. “SAPA: Similarity-Aware Point Affiliation for Feature Upsampling.” ArXiv abs/2209.12866 (2022): n. pag.
>
> [6] Lu, Hao, et al. "FADE: Fusing the assets of decoder and encoder for task-agnostic upsampling." ECCV 2022.

---

> > ### Author Response · Authors · 2023-11-23
> > **Thank you for your review!**
> >
> > We appreciate your thoughtful review of our work. If the rebuttal helped clear up things for you we kindly ask that you increase your score to give FeatUp a fighting chance. Thanks for your consideration!

---

### Official Review · Reviewer_2Fqt · 2023-11-01

**Soundness:** 4 excellent
**Presentation:** 3 good
**Contribution:** 4 excellent
**Rating:** 6
**Confidence:** 4

**Summary:**

Deep features are usually in a low resolution due to the usage of pooling operations. This paper introduces FeatUp to restore the lost spatial information in deep features. FeatUp has two variants: one that guides features with high-resolution signal in a single forward pass, and one that fits an implicit model to a single image to reconstruct features at any resolution. Both approaches use a novel multi-view consistency loss with deep analogies to NeRFs. FeatUp outperforms other feature upsampling approaches in class activation map generation, semantic segmentation, and depth prediction.

**Strengths:**

This paper is well motivated and quite novel. The writing is professional and convincing. The improvement over existing approaches is solid and nontrivial. The proposed method can be used as a drop-in replacement for deep feature upsampling, which could be useful for this community if the claimed improvement can be easily obtained.

**Weaknesses:**

The training details of the FeatUp model when used as a drop-in replacement for existing features are missing. How do you train it? What data do you use for the training? What is the objective and loss for the training?

The experimental details in Table 1 and Table 2 are unclear, making it difficult to reproduce the results. Which networks/models and backbones are used for various downstream tasks? How do you train these models? Are all upsampling approaches evaluated under the same setting?

There are no numeric results for ablation study in this paper, and this is no ablation study in the main part. This paper has many designs and components (Eq. (1) – Eq. (8)), and it is important and necessary to evaluate each of these designs and components. Recently, ablation study is also a necessary part of computer vision papers, especially for deep learning papers. This is the biggest weakness of this paper.

Will the code be released? This is not mentioned in the paper. If the claimed improvement can be easily obtained, the code and pretrained models would be valuable to this community.

**Questions:**

Many details are unclear in the paper. Please see the above weaknesses.

---

> ### Author Response · Authors · 2023-11-18
>
> We thank the reviewer for their helpful comments.
>
> **Additional Experimental Details**
>
> We add all requested additional experimental details to our supplemental material. We provide hyperparameters for both FeatUp variants in Table 7 of the supplement, and provide backbone and training/validation data usage when a new experiment is introduced in the paper. We summarize important details below:
>
> **Experimental details - Table 2 (SegFormer)**
>
> When evaluating FeatUp as a drop-in replacement for upsampling operations in the SegFormer architecture, we use the encoder-decoder architecture and training objective detailed by Xie et al. [1] and followed by Lu et al. [2] for FADE evaluations. Specifically, we train SegFormer on ADE20k [3] (20,210 training images and 2,000 validation images) for 160k steps. To validate that our evaluation setup matches that of existing literature, we also compute FLOPs for SegFormer with various drop-in upsamplers and augment Table 2. These counts are comparable with those in [4], confirming that our architectural setup matches that of existing literature.
>
> **Experimental details - Table 1 (CAM, linear probes)**
>
> We provide details on the datasets, backbones, and feature resolutions for these evaluations in Section 6.12 of the supplement, but have revised the main section to also provide more details. Below, we provide these additional details:
> - CAM scores: Average Drop and Average Increase are computed across 2,000 images from the ImageNet [5] validation set. These images are randomly selected (and kept constant across all evaluations) except for the criteria that the backbone’s classification of the image is equal to the ground truth label (otherwise the measures of Average Drop and Increase would be inaccurate). The FeatUp (JBU) upsamplers are trained on the ImageNet training set for 2,000 steps, and never see the validation images at training time. We use a frozen ViT-S/16 pre-trained in a supervised manner on ImageNet as the featurizer, and extract CAMs by applying a linear classifier after max-pooling ViT features. Upsampling is done (14x14 → 224x224) on the features themselves, and CAMs are obtained from these high-resolution maps.
> - Linear probe (segmentation): We compute mIoU and accuracy over the COCO-Stuff validation set (27 classes) [6]. The FeatUp (JBU) upsamplers are trained on the COCO-Stuff training set for 2,000 steps, and never see the validation images at training time. We use a frozen ViT-S/16 pre-trained in a supervised manner on ImageNet as the featurizer, upsample the resulting features (14x14 → 224x224), and extract 27-channel segmentation maps by applying a linear layer on the features.
> - Linear probe (depth estimation): We compute RMSE and $\delta$ > 1.25 scores over the COCO-Stuff validation set, with monocular depth outputs from the MiDaS v3 hybrid model [7] as ground truth. We use a frozen ViT-S/16 pre-trained in a supervised manner on ImageNet as the featurizer, upsample the resulting features (14x14 → 224x224), and extract 1-channel depth maps by applying a linear layer on the features.
>
> **Quantitative ablations**
>
> We thank the reviewer for bringing up the need for more rigorous evaluation of FeatUp. We have added quantitative ablations of both FeatUp variants on Table 1 (CAM, segmentation linear probe, and depth linear probe) evaluations, and ablations of FeatUp (JBU) on Table 2 (end-to-end SegFormer) evaluations. In all experimental settings, each ablation degrades segmentation performance, with the MLP exclusion generally being the most detrimental. See the global response for more details.
>
> **Code release**
>
> We agree with the reviewer that open-sourced code and pre-trained models are valuable assets to the community. Accordingly, we plan to release the code for both variants of FeatUp, its corresponding training code, and pretrained upsampling modules.
>
> **All details and new results in this response have been added to our paper revision (in yellow highlighted text).**
>
> **References**
>
> [1] Xie, Enze, et al. "SegFormer: Simple and efficient design for semantic segmentation with transformers." NeurIPS 2021.
>
> [2] Lu, Hao, et al. "FADE: Fusing the assets of decoder and encoder for task-agnostic upsampling." ECCV 2022.
>
> [3] Zhou, B., et al. “Semantic understanding of scenes through the ade20k dataset”. IJCV 2019.
>
> [4] Liu, Wenze, et al. "Learning to Upsample by Learning to Sample." ICCV 2023.
>
> [5] Deng, Jia, et al. "Imagenet: A large-scale hierarchical image database." CVPR, 2009.
>
> [6] Caesar, Holger et al.. "Coco-stuff: Thing and stuff classes in context." CVPR. 2018.
>
> [7] Ranftl, René, et al. "Towards robust monocular depth estimation: Mixing datasets for zero-shot cross-dataset transfer." IEEE PAMI.

---

> ### Author Response · Authors · 2023-11-23
> **Thank you for your comments**
>
> We appreciate your thoughtful review of our work. If the rebuttal helped clear up things for you we kindly ask that you increase your score to give FeatUp a fighting chance. Thanks for your consideration!

---

### Author Response · Authors · 2023-11-18

We thank all reviewers for their feedback. We are glad they found our work well-motivated and novel [2Fqt, mrZx] with nontrivial improvements [2Fqt, 2f29, mrZx]. In this response we address shared concerns. **We also updated our draft and highlighted changes in yellow.**

**Quantitative Ablations**

We ablate aspects of FeatUp (Implicit):
- Downsampler (simple vs. attention)
- Presence of outlier detection in loss function
- TV regularization

Additionally we ablate the architecture of FeatUp (JBU):
- No MLP ($MLP$ in Equation 6)
- Replacing the softmax with a gaussian kernel w.r.t. euclidean dist. (original JBU formulation)
- Replacing the softmax with a gaussian kernel w.r.t. cosine dist. (modification of JBU)

We evaluate all variants on all settings from Table 1. Below are the results for the implicit and JBU upsamplers respectively:

|  |  |  | CAM Score |  | Semantic Seg. |  | Depth Estim. |  |
|---|---|:---:|---:|---|---:|---|---:|---|
| **Downsampler** | **O.D.** | **T.V.** | **A.D.** | **A.I.** | **Acc.** | **mIoU** | **RMSE** | **d > 1.25** |
| Attn | Y | Y | **8.84** | **5.60** | **71.58** | **47.37** | **1.04** | **0.927** |
| Simple | Y | Y | 9.07 | 5.06 | 70.95 | 46.79 | 1.11 | 0.916 |
| Attn | N | Y | 8.91 | 5.55 | 71.26 | 46.89 | 1.08 | 0.920 |
| Attn | Y | N | 9.10 | 5.00 | 68.06 | 44.36 | 1.11 | 0.913 |

|  | CAM Score |  | Semantic Seg. |  | Depth Estim. |  |
|:---:|---:|---|---:|---|---:|---|
| **Ablation** | **A.D.** | **A.I.** | **Acc.** | **mIoU** | **RMSE** | **d > 1.25** |
| Original | **9.83** | **5.24** | **68.77** | **43.41** | **1.09** | **0.938** |
| - MLP | 10.04 | 5.10 | 68.12 | 42.99 | 1.14 | 0.917 |
| - Softmax,  + Euclidean | 9.98 | 5.19 | 68.68 | 43.16 | 1.10 | 0.928 |
| - Softmax,  + Cosine | 9.97 | 5.21 | 68.49 | 43.15 | 1.12 | 0.924 |

We also ablate JBU architecture decisions on our end-to-end segmentation training:

|  | Original | No MLP | Euclidean | Cosine |
|---|---|:---:|---:|---|
| **mIoU** | **44.2** | 42.9 | 43.8 | 43.7 |
| **mAcc** | **55.8** | 54.7 | 54.5 | 55.3 |
| **aAcc** | **80.7** | 79.4 | 80.0 | 80.4 |

We find that removing any component hurts performance across all metrics. In particular, removing TV reg. in the implicit network significantly reduces quality - paired with Fig. 9 and 11, we conclude that this is from high-frequency noise. With our JBU variant, removing the MLP (thus reducing expressiveness) degrades performance the most.

**Quantitative FLOP measurements**

We add FLOP counts in Table 2 and analyze scaling in Figure 16; our JBU operator is efficient and competitive with other methods.

**Clarifying the role of the saliency map**

FeatUp's downsampling approximates a network’s pooling process, and is analogous to NeRF's ray-marching approximating the physics of image formation. As seen in Fig. 6, most networks preserve the rough location of objects in their features, leading us to use blur/pooling operators. The simplest of these is avg. pooling, but we can do better by generalizing it to a learned kernel. To map back to NeRF, this is like adding learned lens distortion parameters to the ray-marcher so NeRF can better fit the data.

As shown in Fig. 9 and described in Section 3.1, even a learned kernel cannot capture dynamic receptive fields or object salience. For example if part of an object is in a transformer patch, the whole feature changes. We capture effects like this by making the learned kernel dependent on image content using a 1x1 conv (we don’t need anything bigger than this). Fig. 13 shows that the salience network focuses on certain attributes (e.g. object boundaries, important small objects). We note that many common pooling strategies (e.g. avg. pooling or nearest/bilinear resizing) are special cases of our learnable method.

**Practicality of FeatUp (Implicit)**

We highlight several reasons why implicit features are useful for the community:
- Sometimes quality is more important than speed (e.g. in medicine, geospatial analysis). Understanding a network’s rationale at high res can help practitioners make better decisions. Many existing methods already take this approach and trade speed for quality; for example, [1] takes over 30 minutes per image. Furthermore, methods such as DDIM, NeRF, and Deep Image Priors [2-4] all take more than a single network evaluation.
- They reduce storage of large features by over 100x. This is useful for fine-grained retrieval over large corpora.
- They enable resampling features at arbitrary points. This technique appears in STEGO [5] to remove grid artifacts.
- They enable direct computation of spatial gradients, which can localize object boundaries where features change quickly. This technique appears in [6] for surface normals.
- Implicit networks can learn good sub-pixel representations, similar to DIP-based image super-resolution.

**Improvements to Related Work**

We include all suggested relevant work in the paper. We also add an additional baseline, IFA [7], as requested.

---

> ### Author Response · Authors · 2023-11-18
>
> **References**
>
> [1] Ahn, Jiwoon et al. “Weakly Supervised Learning of Instance Segmentation With Inter-Pixel Relations.” 2019 IEEE/CVF Conference on Computer Vision and Pattern Recognition (CVPR) (2019): 2204-2213.
>
> [2] Song, Jiaming et al. “Denoising Diffusion Implicit Models.” ArXiv abs/2010.02502 (2020): n. pag.
>
> [3] Mildenhall, Ben et al. “NeRF: Representing Scenes as Neural Radiance Fields for View Synthesis.” Commun. ACM 65 (2020): 99-106.
>
> [4] Ulyanov, Dmitry et al. “Deep Image Prior.” International Journal of Computer Vision 128 (2017): 1867 - 1888.
>
> [5] Hamilton, Mark et al. “Unsupervised Semantic Segmentation by Distilling Feature Correspondences.” ArXiv abs/2203.08414 (2022): n. pag.
>
> [6] Sitzmann, Vincent et al. “Light Field Networks: Neural Scene Representations with Single-Evaluation Rendering.” Neural Information Processing Systems (2021).
>
> [7] Hu, Hanzhe et al. “Learning Implicit Feature Alignment Function for Semantic Segmentation.” European Conference on Computer Vision (2022).

---

### Author Response · Authors · 2023-11-21
**Thank you for your reviews**

Thanks to all reviewers for their initial round of thorough feedback that helped us strengthen our work. As the discussion period wraps up, please let us know if there is anything else we can clarify - if we have addressed all concerns, we ask that the reviewers consider raising their scores.

---

### Meta-Review · Area_Chair_PE8o · 2023-12-05

**Metareview:**

This paper presents an approach to retain the spatial information in features from deep learning models that is typically lost due to spatial pooling operations.
The approach, called FeatUp, guides the features with high resolution signals. The authors also show another variant that uses an implicit model to reconstruct features at any resolution. Experiments are conducted on class activation map generation, semantic segmentation, and depth prediction where the method shows solid improvements in performance.
The paper’s overall presentation can be improved significantly — there is a lack of focus in the writing, the experimental details are missing (author response addresses this), missing ablations.

**Justification For Why Not Higher Score:**

The paper presents good results however the presentation quality of this work is lacking. The author rebuttal addresses issues around missing experimental details and ablations. However, the writing updates and refocussed motivation/related work are critical to have a high impact paper.

**Justification For Why Not Lower Score:**

Novel framework for adding spatial information back to deep learning features. The gains on multiple different benchmarks seem solid and robust.

---

### Decision · Program_Chairs · 2024-01-16

Accept (poster)